# Receptor-informed network control theory links LSD and psilocybin to a flattening of the brain's control energy landscape

S. Parker Singleton [1] ✉, Andrea I. Luppi [2,3], Robin L. Carhart-Harris[4,5], Josephine Cruzat [6,7], Leor Roseman[4], David J. Nutt [4], Gustavo Deco [7,8,9,10], Morten L. Kringelbach [11,12,13], Emmanuel A. Stamatakis [2] & Amy Kuceyeski [1,14]

Psychedelics including lysergic acid diethylamide (LSD) and psilocybin temporarily alter subjective experience through their neurochemical effects. Serotonin 2a (5-HT2a) receptor agonism by these compounds is associated with more diverse (entropic) brain activity. We postulate that this increase in entropy may arise in part from a flattening of the brain's control energy landscape, which can be observed using network control theory to quantify the energy required to transition between recurrent brain states. Using brain states derived from existing functional magnetic resonance imaging (fMRI) datasets, we show that LSD and psilocybin reduce control energy required for brain state transitions compared to placebo. Furthermore, across individuals, reduction in control energy correlates with more frequent state transitions and increased entropy of brain state dynamics. Through network control analysis that incorporates the spatial distribution of 5-HT2a receptors (obtained from publicly available positron emission tomography (PET) data under non-drug conditions), we demonstrate an association between the 5-HT2a receptor and reduced control energy. Our findings provide evidence that 5-HT2a receptor agonist compounds allow for more facile state transitions and more temporally diverse brain activity. More broadly, we demonstrate that receptor-informed network control theory can model the impact of neuropharmacological manipulation on brain activity dynamics.

Serotonergic psychedelics like lysergic acid diethylamide (LSD) and psilocybin induce a profound but temporary alteration of perception and subjective experience[1]. Combined with non-invasive neuroimaging such as functional MRI, these drugs offer a unique window into the function of the human mind and brain, making it possible to relate mental phenomena to their neural underpinnings.

A decade of neuroimaging studies has informed novel insights regarding psychedelic action in the brain[2]. One model, known as RElaxed Beliefs Under Psychedelics (REBUS)[3], integrates previous accounts of psychedelic action (the Entropic Brain Hypothesis)[4,5] with the view of the brain as a prediction engine, whereby perception and belief are shaped by both prior knowledge and incoming information. The REBUS model postulates that psychedelics alter conscious experience via their agonist action at serotonin 2a (5-HT2a) receptors, which have especially high expression in higher-order cortical regions. Agonist-induced dysregulation of spontaneous activity in these regions is postulated to translate into decreased precision-weighting on prior beliefs—which has reciprocal (enabling) implications for bottom-up information flow. It is theorized that the observed increase in entropy of brain activity under psychedelics is reflective of reduced

energetic demands or barriers for the brain to navigate its dynamic landscape. However, this hypothesis remains thus far untested.

Understanding and being able to objectively measure the mechanism(s) of psychedelics is paramount if we aim for their therapeutic use in psychiatric or neurologic disorders. Imaging the brains of healthy individuals under the effects of psychedelics offers data from which we can begin to build and test computational, neurobiologically informed models of psychedelic action. Recent work using such approaches has shown that the effects of serotonergic psychedelics on the dynamics of human brain activity are critically dependent on their action at 5-HT2a receptors. Whole-brain neural-mass models have implicated the 5-HT2a receptor distribution across the cortex in shaping brain dynamics under the effects of LSD and psilocybin[6,7].

An alternative computational approach to modeling brain dynamics is network control theory, which focuses on quantifying and controlling how a dynamical system moves through its state space. It is well-known that even at rest the brain is not static, but rather it dynamically alternates between a number of recurrent states[8–15]. Such recurrent brain states may be relevant for cognition[16–19] and even consciousness[20–26], and have been shown to undergo prominent reorganization during the psychedelic state induced by LSD[6,27] and psilocybin[7,28]. Crucially, network control theory approaches enable mapping of the brain's energy landscape by quantifying the energy required to transition between these recurrent states (Fig. 1a, b). This type of energy can be referred to as 'control' or 'transition' energy. Recent work utilized these tools to demonstrate that although the resting human brain has a spontaneous tendency to prefer certain brain state transitions over others, cognitive demands can overcome this tendency in a way that is associated with age and cognitive performance. This work demonstrates that network control theory approaches can reveal neurobiologically and cognitively relevant brain activity dynamics[29–31].

Here, we leverage recent advances in network control theory to probe the relationship between energetic demands and entropy in the psychedelic state: we combine functional MRI data from two previously published experiments comparing drug (LSD[32] or psilocybin[33]) to placebo, with separately obtained (under non-drug conditions) structural (white matter) connectivity from diffusion MRI (dMRI)[34] and receptor density maps from positron emission tomography (PET)[35]. We hypothesized that the energy required to transition between brain states would decrease under LSD and psilocybin compared to placebo and, furthermore, that the amount of control energy reduction would correlate with increases in entropy on an individual level. Finally, we tested the mechanistic hypothesis that serotonergic action at 5-HT2a receptors is responsible for this reduction in transition energy by demonstrating that the specific spatial pattern of 5-HT2a receptor expression flattens the energy landscape more than any other receptor distribution tested (Fig. 1c).

## Results

We analyzed 30 min of resting-state data acquired from 15 subjects over two sessions, either under the influence of LSD or a placebo[32]. To test the generalizability of our findings across datasets and with a different psychedelic compound, we replicated the analysis using 10 min of resting-state data acquired from 9 volunteers over two sessions, either under the influence of psilocybin or a placebo[33]. Importantly, the LSD scans were acquired approximately 2 h following drug/placebo infusion in order to measure the "peak effects" of the drug, whereas the psilocybin participants were scanned immediately after infusion.

### Data-driven clustering of brain activity patterns reveals recurrent states of opposing network activation

Our first step was to identify recurrent states of brain activity. One commonly used approach to identifying recurrent brain states is

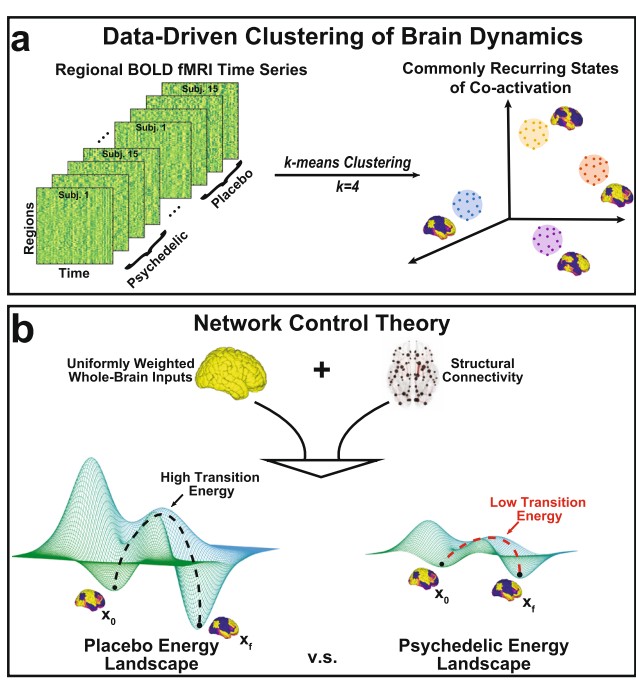

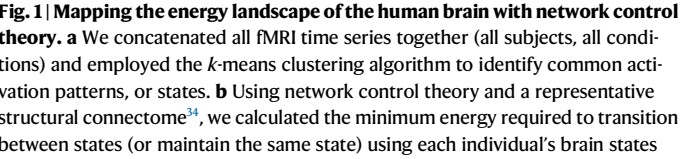

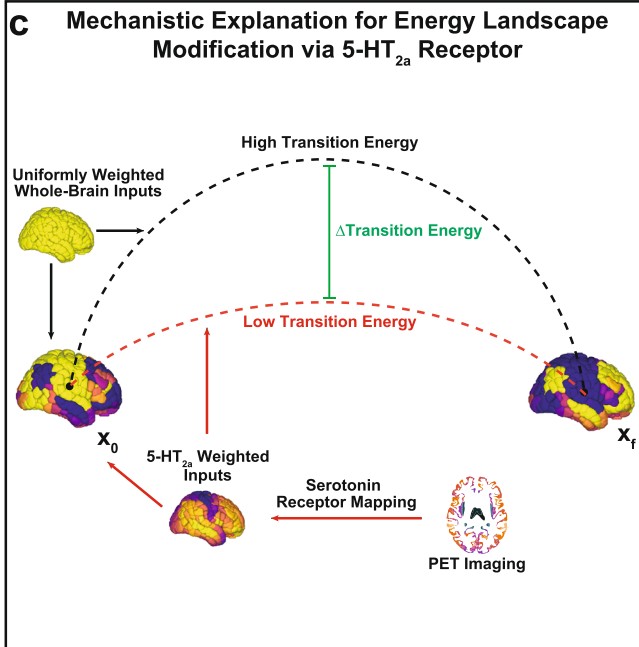

**Fig. 1 | Mapping the energy landscape of the human brain with network control theory. a** We concatenated all fMRI time series together (all subjects, all conditions) and employed the *k*-means clustering algorithm to identify common activation patterns, or states. **b** Using network control theory and a representative structural connectome[34], we calculated the minimum energy required to transition between states (or maintain the same state) using each individual's brain states derived from the psychedelic and placebo conditions separately. Our calculations reveal an energy landscape that is flattened by LSD and psilocybin. **c** Weighting the energy calculations of the placebo brain states with inputs from PET-derived receptor density maps of the serotonin 2a receptor[35] also resulted in a flattened energy landscape, providing a mechanistic explanation for these drug's flattening effects.

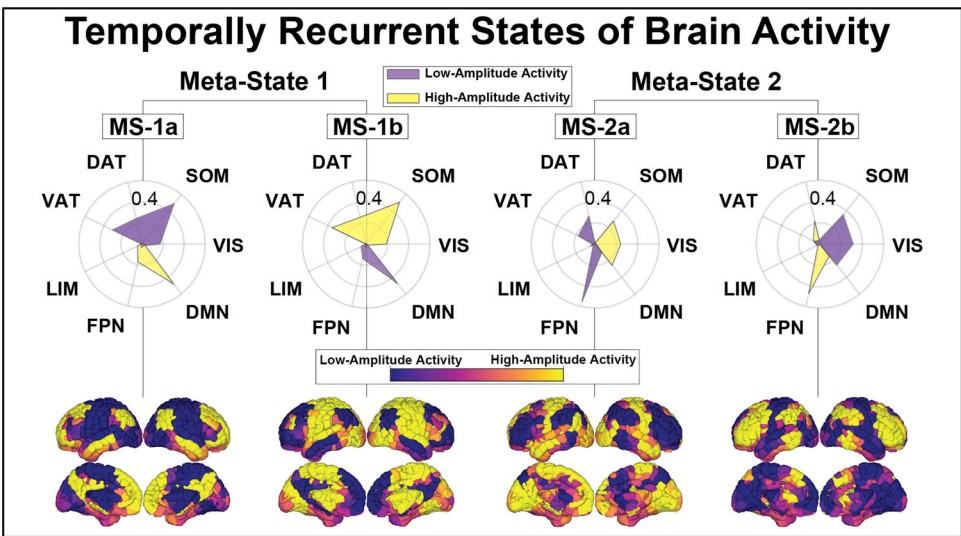

**Fig. 2 | Recurrent states of brain activity.** Group average recurrent brain states are represented by the mean activation pattern across all subjects and conditions for each of the 4 clusters (brain representations at the bottom of the figure). For each brain state, we separately calculated the cosine similarity (radial plots) of its high-amplitude (supra-mean) activity and low-amplitude (sub-mean) activity to a priori resting-state networks[91] (RSNs); resulting similarity measures are represented via radial plots[29]. Meta-State 1 (MS-1) is composed of two sub-states (MS-1a and MS-1b) which are characterized by the contraposition of the somatomotor and ventral attention/salience networks with the default-mode network, whereas the Meta-State 2 (MS-2; MS-2a and MS-2b) is characterized by the contraposition of the default-mode, somatomotor and visual networks with the frontoparietal network. The dichotomy of these states can be observed visually in the radial plots and on the rendered brain volumes, and is confirmed via their negative, significant Pearson correlation (SI Fig. 3b). States shown here are from the LSD dataset. States derived from the psilocybin dataset are highly similar (SI Fig. 4). DAT dorsal attention network, DMN default mode network, FPN frontoparietal network, LIM limbic network, SOM somatomotor network, VAT ventral attention network, and VIS visual network.

through the k-means clustering algorithm[6,7,29,36], whereby brain activation patterns from each individuals' scans are grouped into a pre-specified number of clusters $k$. Here, data-driven clustering of regional activity patterns identified $k = 4$ stable clusters that achieved optimal division of the data (see Materials and Methods: Extraction of brain states for choice of $k$). The four clusters can be divided into two meta-states (Meta-State 1 and Meta-State 2, Fig. 2), each composed of two sub-states that represent opposing activation patterns (MS-1a/b and MS-2a/b, Fig. 2). Dichotomy of the brain's dynamic states has previously been observed[29,37] and is consistent with hierarchical organization[38,39].

### Psychedelics modulate brain dynamics by increasing occupancy in MS-1 and decreasing occupancy in MS-2

To identify the effects of LSD and psilocybin on brain state dynamics, each subject's fMRI data were characterized in terms of the four identified brain states. From each subject's temporal sequence of brain states (Fig. 3a) we obtained a systematic characterization of the temporal dynamics of the 4 states, namely, their fractional occupancies, or the probability of occurrence of each state (Fig. 3b, c, left), dwell times, or the mean duration that a given state was maintained, in seconds (Fig. 3b, c, center), appearance rates, or how often each state appeared per minute (Fig. 3b, c, right), and transition probabilities, or the probability of switching from each state to every other state (Fig. 4a, b, left).

We found that for both psychedelic and placebo conditions, the brain most frequently occupies MS-1 (higher fractional occupancy) whose constituent sub-states are also visited for the longest periods of time (highest dwell times) (Fig. 3b). LSD modifies the fractional occupancy of these states by decreasing the dwell times of MS-2 and further increasing dwell times of the already dominant MS-1 (Fig. 3b). No differences in appearance rate for the 4 sub-states were found when comparing the LSD and placebo conditions. In psilocybin, we found that fractional occupancy shifted in the same direction as LSD, however these changes were not significant. Interestingly, there were no

significant changes in the dwell times under psilocybin, but there were significant increases in the appearance rates of two of the states (Fig. 3c). Possibly, this highlights a subtle difference in the two compounds' impact on brain dynamics or a difference in dynamics under drug onset versus peak effects.

Empirical transition probabilities were calculated independently for each individual and each condition (Fig. 4a, b, left). We note changes during LSD and psilocybin that are consistent with the results observed in Fig. 3. In the LSD data we observe significant increases in the probability of persistence for MS-1 and corresponding decreases in MS-2 (Fig. 4a, left, diagonal). Under psilocybin we observe decreased persistence of MS-2 but not a corresponding increased persistence of MS-1 (Fig. 4b, left, diagonal). Possibly this indicates a difference in the compounds or timeline of administration. In both cases, we see an increased probability of transitioning from states in MS-2 to MS-1 states (Fig. 4a, b, left, off-diagonal).

### Network control theory reveals psychedelic-induced flattening of the brain's control energy landscape

We next sought to provide a direct test of our hypothesis about decreased control energy requirements to transition between different states under psychedelics. To this end, we turned to network control theory[29,30,40–42], which offers a framework to quantify the ease of state transitions in a dynamical system. Specifically, we calculated the transition energy (TE), which is the minimum amount of energy that would need to be injected into a network (here, the structural connectome[34]) to induce transitions between the possible states of its functional dynamics. Note that the transition energy from a given state to itself is the energy required to remain in that state, sometimes referred to as "persistence energy". For each subject and condition, we calculated the energy needed to transition between each pair of brain states. On an individual level, brain states were defined as the centroid of all TRs assigned to each state during that individual's psychedelic or placebo scans. Comparing the two conditions, we found that both LSD and psilocybin

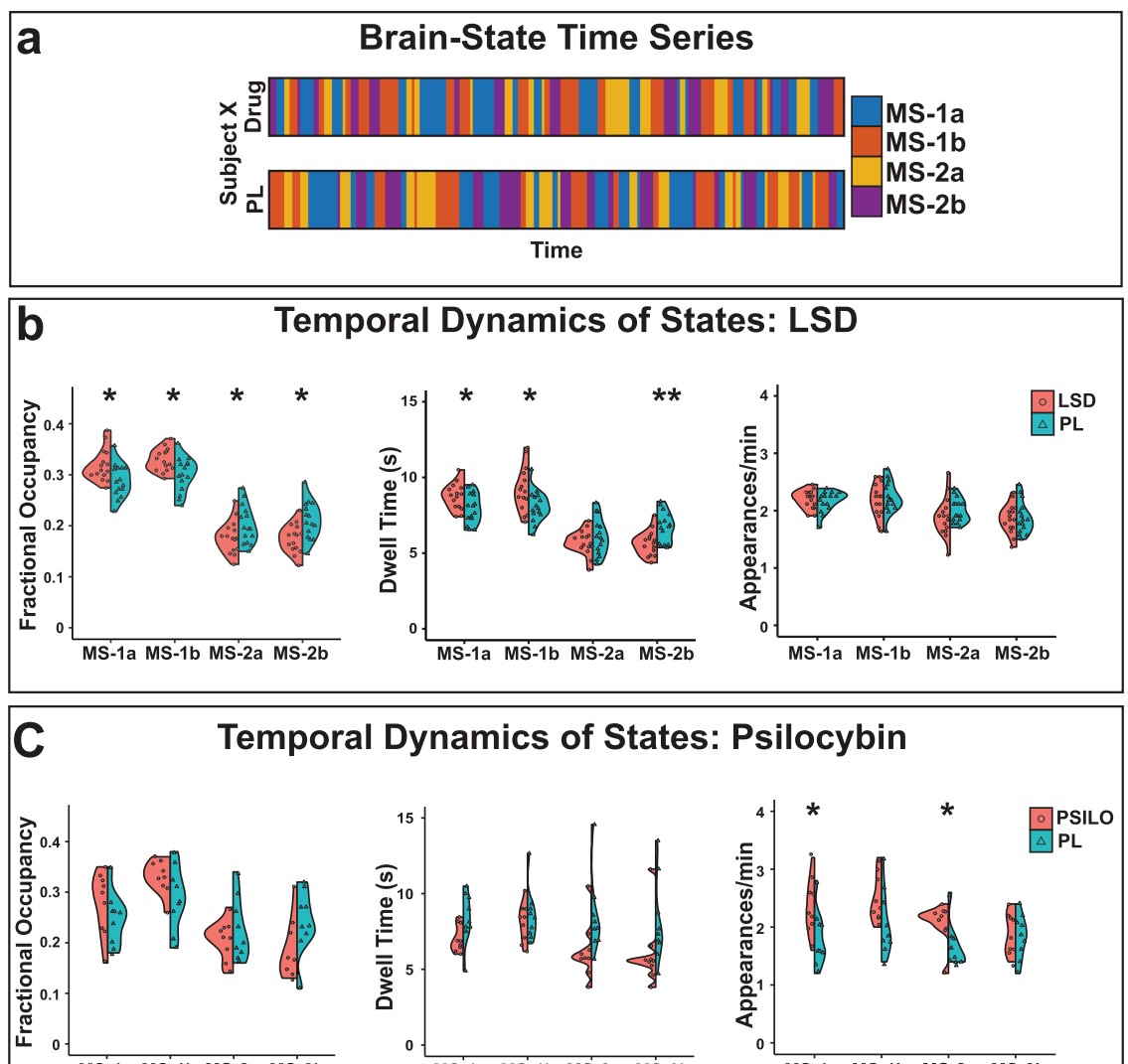

**Fig. 3 | Temporal brain dynamics shift under psychedelics. a** *k*-means clustering of the BOLD time series resulted in a brain state time series for each of the individuals' two scans[29]. We then calculated each brain states' (left) fractional occupancy, (center) average dwell time, and (right) average number of appearances per minute for each individual and condition in the **b** LSD and **c** psilocybin datasets. Comparisons were made using two-sided *t*-tests and *p*-values were corrected for multiple comparisons (Benjamini-Hochberg) where indicated. *uncorrected $p < 0.05$, **corrected $p < 0.05$. Source data are provided as a Source Data file.

lowered the TE (Fig. 4a/b, center left) between all possible combinations of initial and final brain states.

Importantly, network control theory requires a specification of a set of "control points" where energy is injected into the system to induce the desired transition. For the previous analysis, we chose uniform inputs over all brain regions. However, one can also ask whether this effect may be driven by a specific set of regions[29]. This is relevant because the changes in brain function under investigation in the present study arise from either the administration of LSD or psilocybin. The serotonin 2a (5-HT2a) receptor is well established as the site responsible for the key characteristic subjective[43–46] and neural[6,7,47] effects of LSD, psilocybin, and other classic serotonergic psychedelics, and this receptor is not uniformly distributed across the brain[35]. Therefore, we sought to determine if the specific regional distribution of 5HT2a receptors in the brain could correspond to especially suitable control points for inducing a reduction in transition energy.

To test this hypothesis, we utilized a high resolution in vivo atlas of the serotonin receptor 5HT2a derived from PET imaging to extract biologically relevant weights for our model[35]. First, we recalculated the energy matrices for each placebo condition, this time weighting the

energy injected into every region in proportion to its amount of 5-HT2a expression. In every possible transition, we observed that the 5-HT2a-weighted inputs provided lower TE than the uniform inputs (Fig. 4a, b, center right).

However, it could be argued that giving additional control to some regions will result in a lower control energy, regardless of their particular spatial arrangement. To demonstrate that our results are specific to 5-HT2a receptors' spatial distribution across brain regions, we compared the TEs obtained from the true 5-HT2a distribution, versus 10,000 spin permutations that preserve the set of weights and their spatial autocorrelations but not the regions they correspond to[48,49]. The true distribution of 5-HT2a still resulted in significantly lower energies (Fig. 4a, b, right), demonstrating the critical role of the specific regional distribution of 5HT2a receptors for inducing low-energy state transitions such as those empirically observed under the effects of LSD and psilocybin.

In a final demonstration of the specific importance of the 5-HT2a receptor, we investigated the shift in TEs provided by three additional serotonin receptors (5-HT1a, 5-HT1b, 5-HT4) and the serotonin transporter, 5-HTT, all obtained from the same high-resolution PET atlas[35].

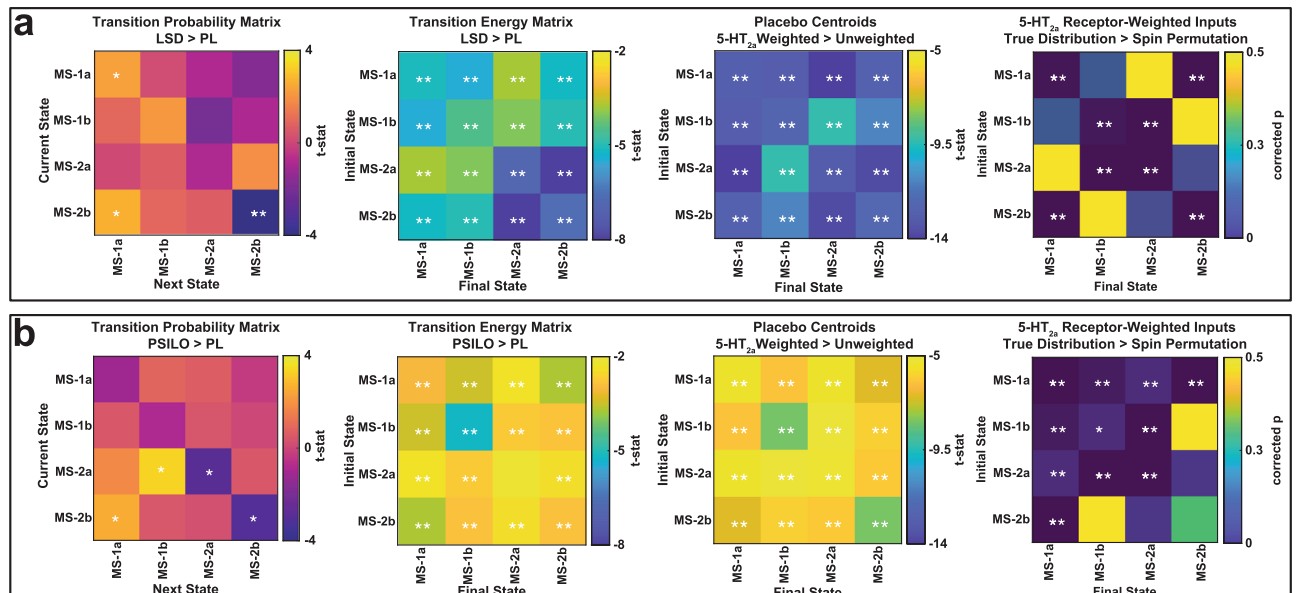

**Fig. 4 | The control energy needed to transition between brain states is reduced by LSD, psilocybin and the spatial distribution of 5-HT2a receptor maps. a** LSD comparisons ($n = 15$) (**b**) Psilocybin comparisons ($n = 9$). (**a**, **b**, left) Comparison of the empirically observed transition probabilities between states, derived from the brain state time series, e.g. Figure 3a. (**a**, **b**, center left) Comparison of the transition energies calculated from placebo brain states versus those calculated from LSD or psilocybin brain states using uniformly-weighted whole-brain inputs. LSD and psilocybin brain states both had significantly lower energy required for every transition. (**a**, **b**, center right) Weighting the control vector by the 5-HT2a receptor density map[35] results in significantly lower energies for the placebo centroids (brain states) compared to uniformly-weighted control vector inputs. (**a**, **b**, right) To

probe the spatial specificity of the previous result, we repeated the calculations using 10,000 spin-permuted receptor maps[48]. We found that the control vector constructed using the true 5-HT2a receptor map resulted in significantly lower energy required for nearly every transition compared to the control vector constructed using the shuffled receptor maps. (See SI for choice of the time-span T over which the transition energy was computed). Comparisons for i-iii were made with two-sided $t$-tests. $P$-values for iv were calculated as the fraction of times that true distribution resulted in higher energies than the spun distribution. $P$-values were corrected for multiple comparisons (Benjamini-Hochberg) where indicated. *uncorrected $p < 0.05$, **corrected $p < 0.05$.

We compared the overall mean of the energy matrix for each individual's 2a-weighted calculations versus all others and found that 5-HT2a was the most effective at lowering the overall energy to transition between empirically defined brain states (Fig. 5). This is especially noteworthy because serotonin 2a receptor agonism plays a prominent role in how LSD, psilocybin, and other classic psychedelics influence neural activity and subjective experience[6,7,43–47]. Together, these results demonstrate that the 5-HT2a receptor is neurobiologically and spatially well-suited for control energy landscape flattening.

Notably, the 5-HT1b receptor resulted in the second largest energy reduction in our model. This is of particular interest as recent animal models have implicated this receptor as the the potential site of antidepressant action of selective serotonin reuptake inhibitors (SSRIs)[50] and may be a route through which serotonergic psychedelics like psilocybin enact their synaptogenesis and potential antidepressant effects[51,52].

**Increased flattening of the control energy landscape is associated with more entropic brain dynamics**
Crucially, the results demonstrating the specific role of 5HT2a receptors in flattening the control energy landscape were based exclusively on calculations using placebo data. Therefore, we next sought to test how the average TE reduction (Fig. 4a, b, center left) may affect empirical transition energies and corresponding brain dynamics. Specifically, we show that, across the 15 individuals, the relative change in control energy induced by LSD was significantly correlated with the empirically observed changes in state dwell times (Fig. 6, left) and appearance rates (Fig. 6, center), $p < 0.05$, uncorrected.

Our results show that the more LSD lowered the average transition energy of a given subject, the more the empirically observed dwell times decreased and the more the empirically observed

appearance rates increased. This is particularly interesting, as there were no group-level differences in appearance rates between the two conditions, and a mix of increased and decreased dwell times depending on the state. While the correlations identified are consistent with our hypothesis of a flattened control energy landscape wherein lower barriers between brain states result in increased frequency of state transitions and shorter state dwell times, it seems this individual-level effect does not translate to a group difference in dwell time or appearance rate (SI Fig. 5a). Interestingly, the psilocybin data had the same directionality of correlation between control energy reductions and decreased dwell time ($r = 0.444$, $p = 0.271$; $n = 9$) and increased appearance rate ($r = −0.453$, $p = 0.259$, $n = 9$) (SI Fig. 4c). However, these data had the additional characteristic of group-level reductions in dwell time and increases in appearance rate under psilocybin compared to placebo that are more straightforward to interpret under the REBUS hypothesis (SI Fig. 5b). Future work with a larger number of subjects may allow further interrogation of this phenomena.

Ratings of the drug's subjective effects were also obtained from each individual (see SI for details) and we hypothesized that transition energy reduction by LSD or psilocybin would also predict a more intense subjective experience. We did not find any significant correlations between energy flattening and subjective ratings; extending the present modeling framework to subjective measures may be a fruitful avenue for future research.

Lastly, we asked whether control energy reduction induced by LSD or psilocybin would correlate with more complex (entropic) brain state time series. This analysis aimed to test the theoretical link between a flatter control energy landscape and more entropic brain activity. One could imagine a scenario where shorter dwell times and larger appearance rates result in a sequence that is highly predictable

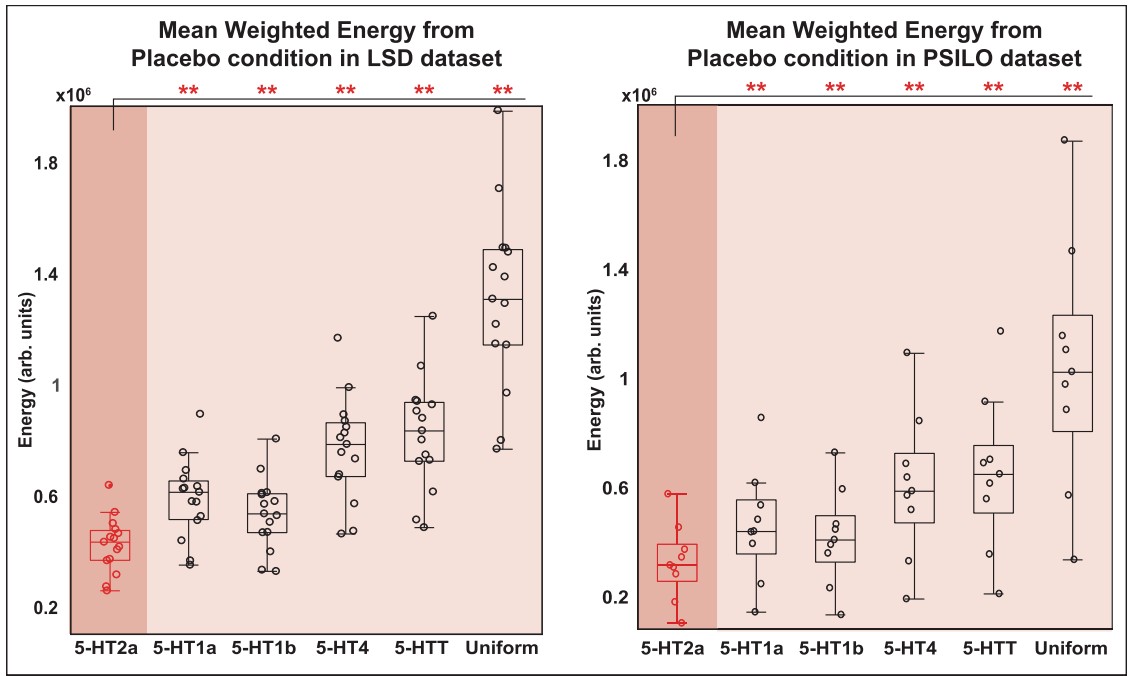

**Fig. 5 | The serotonin 2a receptor flattens the control energy landscape more than any other receptor tested.** We weighted our model with expression maps of other serotonin receptors (5-HT1a, 5-HT1b, and 5-HT4), and the serotonin transporter (5-HTT), and found that 5-HT2a resulted in significantly lower transition energy (averaged across all pairs of states) than all others. (left) Energies calculated from each subject's placebo centroids in the LSD dataset ($n = 15$). (right) Energies calculated from each subject's placebo centroids in the psilocybin dataset ($n = 9$).

On each box, the central line indicates the median, the top and bottom lines refer to the 75th and 25th percentiles, respectively, and the whiskers extend to the most extreme values not considered outliers. Comparisons between 2a energies and all other profiles within each dataset were made using two-sided $t$-tests. $P$-values were corrected for multiple comparisons (Benjamini-Hochberg). **corrected $p < 0.05$. Source data are provided as a Source Data file.

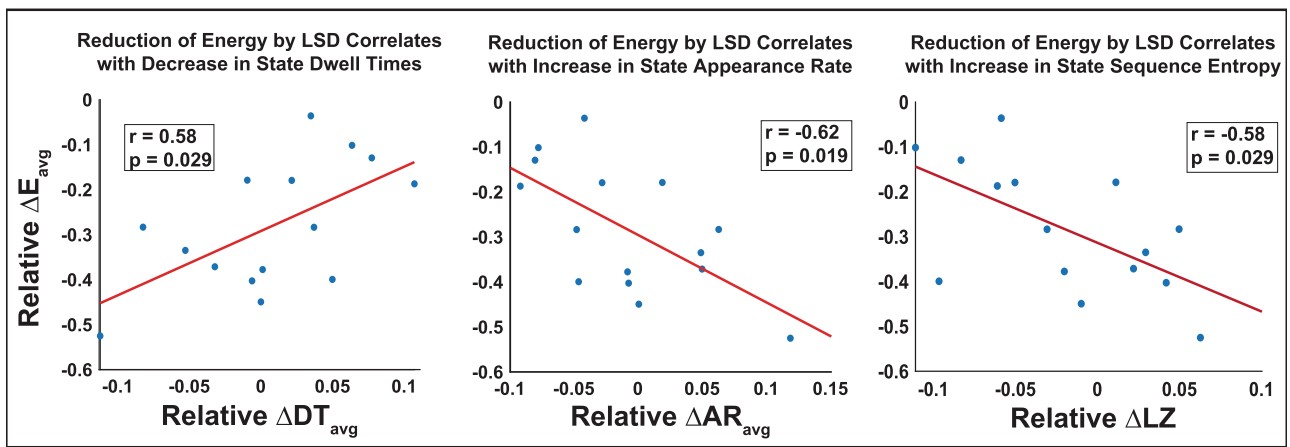

**Fig. 6 | Larger reduction of average transition energy by LSD correlates with more dynamic brain activity across individuals.** Significant Pearson correlations exist between an individual's amount of energy reduction by LSD and the relative change in state (left) dwell times, (center) appearance rates, and (right) entropy of the brain state time series. Relative difference was calculated as $(LSD - PL)/(LSD +$

PL). Partial correlations were calculated while controlling for an individual's head motion (mean framewise displacement). (two-sided, $n = 15$, uncorrected $p$-values). Psilocybin exhibited the same directionality of correlation in each case, however none were significant (SI Fig. 4c). Source data are provided as a Source Data file.

(e.g. [1 2 1 2 1 2]). We wanted to test the hypothesis that the true scenario would be the opposite—namely, that a flatter control energy landscape would in fact correspond to an increase in the diversity of brain dynamics. Numerous studies have linked changes in the entropy of neuroimaging signals to the psychedelic state[5,53–57] and the ability for these compounds to increase neural entropy via 5-HT2a agonism is thought to be a key process in the breakdown of the functional hierarchy of the brain and a central component of REBUS[3,4,58]. To test this hypothesis, we used Lempel-Ziv compressibility to compute the

entropy rate of the temporal sequence of brain meta-states (MS-1 and MS-2). Supporting our hypothesis, we found that the more a subject's energy landscape was flattened by LSD, the more entropic their brain state time series became (Fig. 6, right). Once again, this correlation was not significant in the psilocybin data (SI Fig. 4c); however, we note that the complexity measure deployed here needs particularly large effect sizes to detect differences in such short scans, with few TRs. This result directly and quantitatively links a flatter control energy landscape with more entropic brain activity.

We provide replication of the main findings using data from the LSD study in several ways: (a) analyzing another fMRI scan in which subjects were listening to music (SI Fig. 7), (b) using a different brain atlas (SI Figs. 8 and 9), (c) preprocessing the fMRI data with global signal regression (GSR) (SI Figs. 10 and 11), (d) using a different clustering algorithm (SI Fig. 12), (e) using a wide range of number of clusters k (SI Fig. 14), (f) when clustering each condition (LSD and placebo) separately (SI Fig. 15a, b), (g) and when clustering individuals separately (SI Fig. 15c, d). All results support the main findings.

## Discussion

Here, we combined fMRI data with separately obtained PET and diffusion MRI data under the framework of network control theory to test our hypothesis that serotonergic psychedelics like LSD and psilocybin induce more entropic brain activity in a manner related to a "flattening" of the control energy landscape in the human brain. A flatter energy landscape corresponds to lower barriers to transition between different states of brain activity. This is theorized to correspond to a flattening of the functional hierarchy as well[58], i.e. a relaxation of the weighting of high-level priors—thought to be a pivotal component of psychedelics' therapeutic mechanism[3]. Our results demonstrate: (a) a flattening of the brain's energy landscape, indicated by lower control energy being required to transition between brain states under both LSD and psilocybin, and (b) a correlation between flattening of the energy landscape (reduced energy required for state transitions) and more diverse (entropic) sequences of brain activity under LSD. Combining fMRI with publicly available diffusion MRI and PET information, we were further able to provide computational evidence that (c) the serotonin 2a receptor is especially well-positioned to bring about this flattening of the energy landscape, over and above other 5-HT receptors.

Compared with placebo, subjects in the psychedelic condition spent a larger fraction of time occupying states characterized by the contraposition of the DMN with bottom-up sensorimotor and salience networks (MS-1), and less time in states dominated by the contraposition between DMN and top-down fronto-parietal control network (MS-2) (Fig. 3b). Since our analysis was carried out on resting-state data, it is not surprising that the DMN was prominent across all four brain states[59–62]. It should be noted that this regional-level, activation-based study is not expected to capture the previously reported DMN disintegration[32] (a functional connectivity measure), nor the reduced voxel-level blood flow or BOLD signal indexed 'activity' level of some DMN nodes[33] under psychedelics. Additionally, our quantification of the brain's energy landscape through network control theory revealed that LSD and psilocybin lower the transition energy between all states (Fig. 4a, b, center left). These results are supported by replications using several variations of data, processing and analysis choices.

Given the well-known involvement of 5-HT2a receptors with the neurobiological and subjective effects of classic psychedelics, we next sought to determine if the spatial distribution of 5-HT2a receptors across the human cortex could provide a mechanistic explanation for our results. Weighting the model in proportion to the empirical regional density of 5-HT2a receptors obtained from in vivo PET imaging[35], we found that the resulting transition energies were greatly reduced, mirroring those of the psychedelic condition (Fig. 4a, b, center right). Further, to demonstrate the importance of this receptor's spatial distribution, we tested the true 5-HT2a receptor distribution against a spatial autocorrelation preserving null model (i.e. spin test)[48] and found that the original map consistently resulted in lower energies than the shuffled maps (Fig. 4a, b, right). The calculations were also repeated with other subsets of the 5-HT receptor class, and 5-HT2a receptor was the most effective at reducing energy (Fig. 5), consistent with the known specificity of LSD and psilocybin for this receptor.

The Entropic Brain Hypothesis (EBH)[4,5] proposes that increased neural entropy brought forth by psychedelics is reflected in subjective experience as an increase in the richness or depth of conscious content[63]. We found that at an individual subject level, increased LSD-induced transition energy reductions correlated with more dynamic brain activity (Fig. 6, left, center), thereby relating the theoretical interpretation of transition energy with its role in the empirical de-stabilization of brain state dynamics. We also found that the entropy rate of an individual's sequence of meta-states increased in proportion to the LSD-induced energy reduction (Fig. 6, right), thereby relating the energy landscape of the brain to its entropy.

More broadly, these results demonstrate that the combination of network control theory and specific information about neurobiology (here exemplified by receptor distributions from PET) can offer powerful insights about brain function and how pharmacology may modulate it—opening the avenue for analogous studies on the effects of pharmacological interventions in clinical populations (e.g. depression, schizophrenia)[30,31]. While other recent computational approaches have successfully modeled the effects of serotonergic compounds on dynamic brain states[6,7], the present approach is the first to do so while also quantitatively evaluating the energy landscape of the psychedelic state and its relationship to the entropy of spontaneous neural activity.

Although small sample size is common in neuroimaging studies of psychedelics and other states of altered consciousness due to the inherent difficulties of collecting such data, future replications with larger samples would be appropriate. We also acknowledge that these datasets have been studied extensively before[6,7,27,32,33,57,58,64–68] and replications in different datasets will be warranted to ensure the generalizability of these results. Although the LSD dataset is more suitable for obtaining robust measurements of brain dynamics using the methods presented here, primarily because of the longer scans, and, in addition, the larger sample size, we still found that psilocybin modified the brain's energy landscape and dynamics in a similar fashion to LSD. Future experiments will be necessary to determine if the subtle differences in results found between LSD and psilocybin are a result of sample size, scan length, or pharmacodynamics (different receptor profiles, metabolism, or "peak" versus onset effects).

One other consideration is the objective quality of the clustering—because fMRI measures the smoothly varying hemodynamic response to neuronal activation (BOLD)[69,70], there may be suboptimal separation of clusters. For example, even if neuronal activity does switch between states near-instantaneously, there will still be a smoothing of the measured activity in the fMRI which would result in some TRs occurring at the boundaries of the states. Despite this potential limitation, our current work and previous studies on dynamic states from fMRI have revealed physiologically and behaviorally meaningful cluster-based metrics[7,16,27–29,71–73].

As used here, the term "energy" denotes the magnitude of the input that needs to be injected into the system (the brain's structural connectome) in order to obtain the desired state transition. It should not be confused with metabolic energy of ATP molecules, nor with the energy quantified through connectome harmonic decomposition, which has also been investigated in the context of the LSD dataset[64,67] and other states of altered consciousness[65]. It is also not a direct measure of the brain's variational free energy landscape[74], an information theoretic topic foundational to REBUS. We hypothesize that the empirical changes in control energy demonstrated here indicate a flattening of the posterior, which may or may not arise from a relaxation of prior beliefs and a flattening of the free energy landscape as posited by REBUS. Future work will be required to determine if any direct relationship exists between free energy and control energy.

Additionally, we had hypothesized that the transition energy modifications by LSD would correlate with our participants' subjective experience as captured by intra-scanner visual analog scale ratings, and the 11-factor states of consciousness (ASC) questionnaire[75,76] taken at the end of the day. There may be numerous factors that limit our ability to model these effects. Psychedelics have been found to impair

some aspects of memory recollection in humans[77] (though not autobiographical memory recollection[78]), which may arguably impact on the fidelity of post-hoc subjective ratings and how they correlate with acute, objective brain measures. In addition, both subjective experience ratings and the relative energy landscape (baseline or drug) may be impacted by each individual's prior psychedelic use, individual differences in pharmacological dose response, as well as their own unique structural connectome and 5-HT2a receptor distribution. Indeed, the structural connectome and the PET data used in our analysis were representative examples obtained from population averages (not under drug conditions), rather than unique data derived from each individual in our study. Although these measures are thought to be less variable across individuals than brain activity dynamics, future work could explore how individual differences in the structural connectome or receptor maps influence the energy landscape—and possibly subjective experiences.

Finally, our approach is based on network control theory, which differs from other recent computational investigations using e.g. whole-brain simulation through dynamic mean-field modeling of brain activity[6,7,22]. These latter approaches employ a neurobiologically realistic model of brain activity based on mean-field reduction of spiking neurons into excitatory and inhibitory populations, and have been used to account for non-linear effects of 5HT2a receptor neuromodulation induced by psychedelics. In contrast, network control theory relies on a simpler linear model, which we employed due to its ability to address our prediction that transition (control) energies would be reduced under psychedelics. Additionally, recent evidence suggests that most of the fMRI signal may be treated as linear[79,80]. By using the same linear equation for modeling both the psychedelic and placebo states, we quantify relative changes in the energy landscape while keeping the dynamics of the model the same for each case; implicitly, this also means that we assume that the underlying structural connectivity is providing the same contribution in both conditions. However, using different models for the two cases may represent an avenue for future research. Combining both approaches to capitalize on the strengths of each will be a fruitful avenue for future work.

For the first time, we apply a framework for receptor-informed network control theory to understand how the serotonergic psychedelics LSD and psilocybin influence human brain function. Combining fMRI, diffusion MRI, PET and network control theory, we present evidence supporting the hypothesis that psychedelics flatten the brain's energy landscape and, furthermore, provide a mechanistic explanation for this observed energy reduction by demonstrating that the empirical spatial distribution of 5-HT2a receptor expression is particularly well-suited to flatten the brain activity landscape. This work highlights the potential of receptor-informed network control theory to allow insights into pharmacological modulation of brain function.

## Methods

### Ethics and approval

The original studies were approved by the National Research Ethics Service committee London-West London and conducted in accordance with the revised declaration of Helsinki (2000), the International Committee on Harmonization Good Clinical Practice guidelines, and National Health Service Research Governance Framework. Imperial College London sponsored the research, which was conducted under a Home Office license for research with schedule 1 drugs. For both studies, healthy participants were recruited via word of mouth and provided written informed consent to participate after study briefing and screening for physical and mental health.

### Data collection: LSD

Data acquisition is described in detail previously[32]. In brief, twenty healthy volunteers underwent two MRI scanning sessions at least

14 days apart. A fully randomized, double-blind design is often considered the gold standard; however, experimental blinding is known to be ineffective in studies with conspicuous interventions. Thus, a single-blind, balanced-order design with an inert placebo (offering the simplest and "cleanest" possible control condition) was considered an effective compromise. On one day, participants were given placebo (10 mL saline), and on the other day they received LSD (75 µg in 10 mL saline), infused over two minutes, approximately 2 h before resting-state scanning. Post-infusion, subjects had a brief acclamation period in a mock fMRI scanner. On each scanning day, three 7:20 min eyes-closed resting-state scans were acquired. The first and third scan had no stimulation, while the second scan involved listening to music; this scan was not used in this analysis as we were interested in dynamics in the absence of external stimulation. BOLD fMRI was acquired at 3 T with TR/TE = 2000/35 ms, FoV = 220 mm, 64 × 64 acquisition matrix, parallel acceleration factor = 2, 90 flip angle. Thirty-five oblique axial slices were acquired in an interleaved fashion, each 3.4 mm thick with zero slice gap (3.4 mm isotropic voxels). One subject was excluded due to anxiety, and 4 due to excessive head motion, leaving 15 subjects (four women; mean age, 30.5 ± 8.0) for analysis. Principally, motion was measured using framewise displacement (FD). The criterion for exclusion was subjects with >15% scrubbed volumes when the scrubbing threshold is FD = 0.5. After discarding these subjects we reduced the threshold to FD = 0.4. The between-condition difference in mean FD for the 4 subjects that were discarded was 0.323 ± 0.254 and for the 15 subjects (four women; mean age, 30.5 ± 8.0) that were used in the analysis the difference in mean FD was 0.046 ± 0.032 ($p = 0.0002$).

### Data collection: psilocybin

Data acquisition is described in detail previously[33]. In brief, fifteen health volunteers underwent two MRI scanning sessions at least 14 days apart. In each session, subjects were injected. with either psilocybin (2 mg dissolved in 10 mL of saline, 60-s i.v. injection) or a placebo (10 mL of saline, 60-s i.v. injection) in a counterbalanced design. The infusions began exactly 6 min after the start of the 12-min fMRI scans and lasted 60 s. The subjective effects of psilocybin were felt almost immediately after injection and sustained for the remainder of the scanning session. The 5 min of post-infusion data were used for the present analysis. BOLD-weighted fMRI data were acquired at 3 T using a gradient echo EPI sequence, TR/TE 3000/35 ms, field-of-view = 192 mm, 64 × 64 acquisition matrix, parallel acceleration factor = 2, 90° flip angle. Fifty-three oblique axial slices were acquired in an interleaved fashion, each 3 mm thick with zero slice gap (3 × 3 × 3-mm voxels). Following the same exclusion criteria for motion described above, nine subjects (two women; mean age, 32 ± 8.9) were kept for analysis.

### Data preprocessing

Data pre-processing utilized AFNI, Freesurfer, FSL and in-house code[32,33]. Steps included (1) removal of first three volumes; (2) despiking; (3) slice time correction; (4) motion correction (Please see *SI: Motion* for additional steps taken to account for motion); (5) brain extraction; (6) rigid body registration to anatomical scans; (7) nonlinear registration to 2 mm MNI space; (8) scrubbing - using a framewise displacement threshold of 0.4. The maximum number of scrubbed volumes per scan was 7.1%; scrubbed volumes were replaced with the mean of the preceding and following volumes; (9) spatial smoothing; (10) band-pass filtering (0.01 to 0.08 Hz); (11) de-trending; (12) regression out of 6 motion-related and 3 anatomical-related nuisance regressors. Lastly, time series for 462 gray matter regions[81] were extracted (Lausanne scale 4, sans brain-stem). Regional time-series were de-meaned prior to analysis.

## Structural connectivity network construction

Since diffusion MRI was not acquired as part of the LSD study, the structural connectome used for network control theory analysis was identical to the one used in prior work[27]. Namely, we relied on diffusion data from the Human Connectome Project (HCP, http://www.humanconnectome.org/), specifically from 1021 subjects in the 1200-subject release[82]. A population-average structural connectome was constructed and made publicly available by Yeh and colleagues in the following way[34]. Multishell diffusion MRI was acquired using b-values of 1000, 2000, 3000 s/mm², each with 90 directions and 1.25 mm iso-voxel resolution Following previous work[27,83], we used the QSDR algorithm[84] implemented in DSI Studio (http://dsi-studio.labsolver.org) to coregister the diffusion data to MNI space, using previously adopted parameters[83]. Deterministic tractography with DSI Studio's modified FACT algorithm[85] then generated 1,000,000 streamlines, using the same parameters as in prior work[27,41,83], specifically, angular cutoff of 55∘, step size of 1.0 mm, minimum length of 10 mm, maximum length of 400 mm, spin density function smoothing of 0.0, and a QA threshold determined by DWI signal in the CSF. Each of the streamlines generated was screened for its termination location using an automatically generated white matter mask, to eliminate streamlines with premature termination in the white matter. Entries in the structural connectome $A_{ij}$ were constructed by counting the number of streamlines connecting every pair of regions $i$ and $j$ in the Lausanne-463[81] (sans brain-stem) and augmented Schaefer-232 atlas[86,87] as done previously[27,83]. Lastly, streamline count was normalized by the number of voxels contained in each pair of regions.

## 5-HT receptor mapping

Details for obtaining the serotonin receptor density distribution have been previously described[35] however we provide a brief summary here. PET data for 210 participants (not under the influence of psychedelics) were acquired on a Siemens HRRT scanner operating in 3D acquisition mode with an approximate in-plane resolution of 2 mm (1.4 mm in the center of the field of view and 2.4 mm in cortex)[88]. Scan time and frame length were designed according to the radiotracer characteristics. For details on MRI acquisition parameters, which were used to coregister the data to a common atlas, see Knudsen et al.[89]. The voxelwise average density (Bmax) maps for each receptor were parcellated into regions of interest the Lausanne[81] and augmented Schaefer[86,87] atlases.

## Extraction of brain states

Following Cornblath et al.[29], all subjects' fMRI time series for both conditions were concatenated in time and $k$-means clustering was applied to identify clusters of brain activation patterns, or states. Pearson correlation was used as the distance metric and clustering was repeated 50 times with different random initializations before choosing the solution with the best separation of the data. To further assess the stability of clustering and ensure our partitions were reliable, we independently repeated this process 10 times and compared the adjusted mutual information (AMI)[90] between each of the 10 resulting partitions. The partition which shared the greatest total AMI with all other partitions was selected for further analysis. In general, we found that the mutual information shared between partitions was quite high, suggesting consistent clustering across independent runs (see SI: Assessing the stability of clustering). We chose the number of clusters $k$ via the elbow criterion, i.e. by plotting the variance explained by clustering for $k = 2$ through 14 and identifying the "elbow" of the plot, which was between 4–6 across the various partitions. In addition, increasing $k$ beyond $k = 5$ resulted in a gain of less than 1% of variance explained by clustering, a threshold used previously for determining $k$ (see SI: Choosing k)[29]. We chose $k = 4$ for its straightforward and symmetric interpretation, however the energy landscape findings are replicated with $k = 2$–14 and all findings for $k = 5$ are provided in the Supplementary Information.

## Characterization of brain states and their hierarchy

Each cluster centroid was characterized by the cosine similarity between it and binary representations of seven a priori defined RSNs[29,91] as shown in the radial plots of Fig. 2. Because the mean signal from each scan's regional time series was removed during bandpass filtering, positive values in the centroid reflect activation above the mean (high-amplitude) and negative values reflect activation below the mean (low-amplitude). To quantify the hierarchical relationship between centroids observed in the radial plots, we calculated the Pearson correlation values between all centroids (SI Fig. 3b) and grouped the anti-correlated pairs together, and refer to each individual centroid as a sub-state and the pair collectively as a meta-state[29,37]. For replications with other data or processing choices, the states were ordered and labeled based on their maximum correlation with the original 4 centroids in Fig. 2.

We can extract (1) group-average centroids by taking the mean of all TR's assigned to each cluster (all subjects, all conditions) (shown in Figs. 2), (2) condition-average centroids by taking the mean of all TR's assigned to each cluster separately for each condition (shown in SI Fig. 6; placebo condition-average centroids were used for Fig. 4a, b, right and Fig. 5 calculations), and (3) individual condition-specific centroids by taking the mean of all TRs assigned to each cluster for a single subject and condition (used for Fig. 4a, center left & center right calculations). When taking condition-average centroids (LSD and PL), we find that these two sets of centroids are highly correlated with one another (SI Fig. 6d), and thus are also very similar to the group-average centroids shown here.

## Temporal brain state dynamics

We then analyzed the temporal dynamics of these brain states to observe how they change after administration of LSD and psilocybin[29]. The fractional occupancy of each state was determined by the number of TRs assigned to each cluster divided by the total number of TRs. Dwell time was calculated by averaging the length of time spent in a cluster once transitioning to it. Appearance rate was calculated as the total number of times a state was transitioned into per minute. Transition probability values were obtained by calculating the probability that any given state $i$ was followed by state $j$.

## Energy calculations

Network control theory allows us to probe the constraints of white-matter connectivity on dynamic brain activity, and to calculate the minimum energy required for the brain to transition from one activation pattern to another[29,40,92]. Here, we utilized network control theory to understand the structural and energetic relationships between these states and the 5-HT2a receptor distribution. While this procedure has been detailed elsewhere[29], we summarize briefly here and in the Supplemental Information. We obtained a representative NxN structural connectome $A$ obtained as described above using deterministic tractography from HCP subjects (*see Methods and Materials; Structural Connectivity Network Construction*), where N is the number of regions in our atlas. We then employ a linear time-invariant model:

$$\dot{x}(t) = Ax(t) + Bu(t) \tag{1}$$

where x is a vector of length N containing the regional activity at time $t$. $B$ is an NxN matrix that contains the control input weights. $B$ is the identity matrix for uniform inputs and contains the regional receptor density information in the diagonal when incorporating the 5-HT receptor maps. For the latter case, the diagonal of $B$ was set to 1 plus the normalized regional receptor density value, resulting in a diagonal matrix whose non-zero entries were between 1 and 2. This computational approach allows us to compute the transition energy as the

minimum energy required to transition between all pairs of the substates.

The energy calculations in Fig. 4a, b (center left) consisted of separate calculations for each individual's drug and placebo centroids separately, (center right) utilized each individual's placebo centroids while varying the control input weights *B*, and (right) used the group average placebo centroids, and *B* was varied for each random permutation. Figure 5 again used each individual's placebo centroids, while varying control input weights *B*.

### Lempel-Ziv complexity
In order to quantify the entropy of each subject's brain state time series, we chose the widely used Lempel-Ziv algorithm[93,94]; this algorithm assesses the complexity of a binary sequence in terms of the number of unique patterns it contains. A sequence that contains a larger number of unique patterns is more diverse, making it less predictable and therefore more entropic. The normalized Lempel-Ziv complexity (also known as Lempel-Ziv compressibility) is then the number of patterns found in the sequence, divided by the total length of the sequence. In order to apply this algorithm to our brain state time series, we first had to convert them to binary sequences that returned 0 or 1 for each time point. To do so, we considered the natural grouping of our 4 brain states into two meta-states (Meta-State 1 and Meta-State 2). We consider this simplification to be justified by the fact that direct transitions between sub-states (e.g. MS-1a to MS-1b) were extremely rare (SI Fig. 16a), thereby allowing us to reduce the 4-state time series to a 2-state time series while losing very little information regarding transitions.

### Statistical comparisons
The 5-HT$_{2a}$ - weighted inputs from the true receptor distribution were compared to the randomly shuffled distributions via a permutation test where the true receptor distribution was spin-permuted and the energy matrix re-calculated 10,000 times[48]. *P*-values were calculated as the fraction of times that the randomized distribution resulted in a lower energy than the true distribution. All other metric comparisons were achieved using a two-sided paired *t*-test of group means and were corrected for multiple comparisons with Benjamini-Hochberg where correction is indicated.

### Citation and gender diversity statement
Recent work in neuroscience and other fields has identified a bias in citation practices such that papers from women and other minorities are under-cited relative to the number of such papers in the field[95–99]. Here, we sought to proactively consider choosing references that reflect the diversity of the field in thought, form of contribution, gender, and other factors. We used classification of gender based on the first names of the first and last authors[96,100], with possible combinations including male/male, male/female, female/male, and female/female. Excluding self-citations to the first and last authors of our current paper, the references contain 58.82% male/male, 15.29% male/female, 18.82% female/male, and 7.06% female/female. We look forward to future work that could help us to better understand how to support equitable practices in science.

### Reporting summary
Further information on research design is available in the Nature Research Reporting Summary linked to this article.

## Data availability
The fMRI LSD data are freely available at https://openneuro.org/datasets/ds003059/versions/1.0.0. The voxelwise receptor density maps are freely available at: https://xtra.nru.dk/FS5ht-atlas/. All other functional and structural data, along with parcellated receptor maps and data to reproduce the main figures are available on our Zenodo repository[101]. Source data are provided with this paper.

## Code availability
This project used open-source code cited in the main text, as well as code published by Cornblath et al.[29] and was carried out using MATLAB R2017a and R 3.2.5. A code repository for reproducing the analysis is available on Zenodo[101].

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

## Acknowledgements

S.P.S. is supported by the National Science Foundation Graduate Research Fellowship (Grant No. DGE-1650441). A.I.L. is supported by the Gates Cambridge Trust. R.L.C.-H. was supported by the Alex Mosley Charitable Trust and supporters of the Centre for Psychedelic Research during the period of data collection, analysis and write-up and now holds the Ralph Metzner Distinguished Professorship at UCSF. The original study received support from a Crowd Funding Campaign and the Beckley Foundation, as part of the Beckley-Imperial Research Programme. J.C. is supported by the Spanish Ministry of Science and Innovation under the fellowship BES-2017-080364. D.J.N. is the Edmond J. Safra Prof of Neuropsychopharmacology at Imperial College London. G.D. is supported by the Spanish Research Project (ref. PID2019-105772GB-I00 AEI FEDER EU), funded by the Spanish Ministry of Science, Innovation and Universities (MCIU), State Research Agency (AEI) and European Regional Development Funds (FEDER); HBP SGA3 Human Brain Project Specific Grant Agreement 3 (Grant Agreement No. 945539), funded by the EU H2020 FET Flagship program and SGR Research Support Group support (ref. 2017 SGR 1545), funded by the Catalan Agency for Management of University and Research Grants (AGAUR). M.L.K. is supported by the Center for Music in the Brain, funded by the Danish National Research Foundation (DNRF117), and the Centre for Eudaimonia and Human Flourishing, funded by the Pettit and Carlsberg Foundations. EAS is supported by the Stephen Erskine Fellowship from Queens' College, Cambridge and the Canadian Institute for Advanced Research (L'Institut Canadien de Recherches Avancées; RCZB/072 RG93193). A.K. is supported by the National Institutes of Health (RF1MH123232, R01NS102646 and R21NS104634).

## Author contributions

R.L.C.-H. and D.J.N. obtained funding for, designed, and implemented the original LSD and psilocybin experiments. L.R. assisted in the original studies and preprocessed the fMRI data for this analysis. S.P.S., A.K., J.C., G.D., and M.L.K. conceptualized the present analysis. S.P.S., A.K., & A.I.L. designed and implemented the analytical framework. A.K., E.A.S., R.L.C.-H., G.D., & M.L.K. provided supervision of the present analysis. S.P.S. produced all figures. S.P.S., A.I.L., A.K., and E.A.S. drafted the manuscript, and all authors reviewed, edited, and approved the final version.

## Competing interests

R.L.C.-H. is an advisor to Beckley Psytech, Entheon Biomedical, TRYP Therapeutics, Mydecine, Usona Institute, Synthesis Institute, Osmind, Maya Health, and Journey Collab. D.J.N. is an advisory to COMPAS-SPathways, Neural Therapeutics, and Algernon Pharmaceuticals. All other authors declare no competing interests.

## Additional information

[1]Department of Computational Biology, Cornell University, Ithaca, NY, USA. [2]Division of Anesthesia, School of Clinical Medicine, University of Cambridge, Cambridge, UK. [3]Department of Clinical Neurosciences, University of Cambridge, Cambridge, UK. [4]Center for Psychedelic Research, Department of Brain Science, Imperial College London, London, UK. [5]Psychedelics Division, Neuroscape, University of California San Francisco, San Francisco, CA, USA. [6]Latin American Brain Health Institute (BrainLat), Universidad Adolfo Ibanez, Santiago, Chile. [7]Center for Brain and Cognition, Computational Neuroscience Group, Department of Information and Communication Technologies, Universitat Pompeu Fabra, Roc Boronat 138, Barcelona, Spain. [8]Institució Catalana de la Recerca i Estudis Avançats (ICREA), Passeig Lluís Companys 23, Barcelona, Spain. [9]Department of Neuropsychology, Max Planck Institute for Human Cognitive and Brain Sciences, Leipzig, Germany. [10]School of Psychological Sciences, Monash University, Melbourne, Clayton, VIC, Australia. [11]Department of Psychiatry, University of Oxford, Oxford, UK. [12]Center of Music in the Brain (MIB), Clinical Medicine, Aarhus University, Aarhus, Denmark. [13]Centre for Eudaimonia and Human Flourishing, University of Oxford, Oxford, UK. [14]Department of Radiology, Weill Cornell Medicine, New York, NY, USA. ✉e-mail: sps253@cornell.edu

