## [Peer Review File · Nature Communications]

Receptor-informed network control theory links LSD and psilocybin to a flattening of the brain's control energy landscapeReviewers' comments:

Reviewer #1 (Remarks to the Author):

LSD flattens the brain's energy landscape: evidence from receptor-informed network control theory

In this article the authors use network control theory to explore the effects of psychedelics on explorations of state space of brain activity. The principal finding is that the space is "flattened", i.e. there exists a reduction in the energy needed to move from one state to another. This topic is timely and the results compelling. However, there are many methodological concerns that need to be addressed through additional analyses. In general, I have concerns about the robustness of results with respect to reasonable variation in analysis and processing pipelines.

Comments:

1. The authors assume that a single dynamical system describes the evolution of brain activity irrespective of whether the brain is in a psychedelic state or not. Under these dynamics, the energy appears reduced in a psychedelic state. But maybe the actual effect is a change to the dynamics themselves and therefore the optimal trajectories (and their energies) during psychedelic states are incorrect. In fact, the authors own results seem to support this alternative hypothesis – e.g. the increased entropy of brain dynamics.

Minor comments:

1. A few concerns about the extraction of brain states. The authors justify the solution of $k = 4$ and show that their main findings are replicate with $k = 5$. Do results hold for other k values or just these two? Simply showing the solutions (states) is not sufficient. To make the claim that LSD flattens an energy landscape, the authors need to show that their results are more general and not sensitive on parameterization.
2. Another concern with k-means clustering. Like many other algorithms, k-means will *always* find a k-partition of a dataset, even if no true clusters exist. Can the authors demonstrate that the states they report are, in fact, separated from one another and that the algorithm hasn't just partitioned into clusters what is essentially a continuous space of activity patterns? At the very least, the authors should test other clustering algorithms to verify that their results are not dependent on the k-means algorithm.
3. Another concern about brain states: how does global signal regression impact these results? From the methods section it seems that global signal was not regressed out. Given the contentious nature of this procedure, the authors should verify that global signal regression doesn't impact their main results.
4. It might be useful to confirm that the brain state estimates are equally applicable for all subjects. One way to do this is to treat each frame as an instant of a state, compute the distance from the state centroid and the actual pattern of activity, and measure the error (e.g. calculate difference between every region's activity, square these values, and add them up). Then take the sum over the entire time series. Are there systematic errors where some subjects are not well-represented by the clusters? Are there differences between the two conditions (LSD/PL)? This is an important point: if the states are more representative of some subjects and less representative of others, then this propagates to the estimates of control energy. The authors need to confirm that the differences in control energies are not driven by differences in how well the states represent an individual.

5. Are there differences in motion/data quality between PL and LSD conditions? How are these differences dealt with?
6. It is well known that fMRI data is highly autocorrelated, which might explain self-transitions (state X -> state X). Would results hold if the authors treated every temporally contiguous sequence of state X as a single instance of that state? This effectively means that the probability of self-transitions is zero.
7. The authors permute the spatial positions of weights for the 5-HT2a distribution. This is not appropriate, because there exist spatial autocorrelations in the data that are not preserved in the permuted data. The authors should retest this effect using “spin tests” instead (see Markello & Misic 2021).
8. Weighting structural connections based on streamline counts is not appropriate. All things being equal, larger parcels will tend to be associated with a greater number of streamlines. The authors need to perform some kind of correction in which streamline counts are normalized by some measure of parcel size, e.g. surface area or volume.

References:

Markello, R. D., & Misic, B. (2021). Comparing spatial null models for brain maps. *NeuroImage*, 236, 118052.

Reviewer #3 (Remarks to the Author):

In another reanalysis alongside the 15+ other reanalyses of this dataset (more publications than participants), Singleton and colleagues provide partial evidence for the REBUS model. Although this may be one of the better tests of REBUS, the salami slicing has gotten out of control. Not only is the small sample (N = 15) not independent of all the other tests of REBUS, the analyses themselves are not necessarily independent (e.g., <https://doi.org/10.1016/j.neuroimage.2020.117049> in which similar complexity measures were applied to these data). Moreover, whereas some of this group's past work has occasionally included their psilocybin dataset (e.g., both included in a single publication <https://doi.org/10.1016/j.neuroimage.2020.117049> and <https://doi.org/10.1016/j.cub.2016.02.010>; salami sliced into two separate publications <https://doi.org/10.1038/s41598-017-17546-0> and <https://doi.org/10.1016/bs.pbr.2018.08.009>; see also <https://doi.org/10.1016/j.neuroimage.2019.05.060> for analysis of only the psilocybin dataset when multiple publications of the LSD dataset had already been out), the same analyses to the psilocybin dataset are conspicuously missing (as they tend to be from most of the reanalyses). If I am not mistaken, this group now also has a DMT dataset that together could essentially triple their sample size and provide some generalizability to these findings. REBUS is supposed to be an all-inclusive model of psychedelic (i.e., 5-HT2A) drug action rather than specific to LSD. Additionally, if what is found here is thought to be a general principle of psychedelic drug action on the brain, then the listening to music condition should also be expected to exhibit such effects. Although this paper should not be published in *Nature Communications* (or anywhere until all other datasets are integrated), wherever it may end up, the following points must be considered:

- 1) The abstract needs to state that this is another reanalysis. This point is generally skipped across all of the reanalyses, giving the illusion of independent samples. A trainee recently questioned whether all of Imperial's fMRI studies were actually based on only two datasets of 15 subjects, and although such information is typically found in the methods, many people

simply read abstracts and do not have ties to the researchers themselves to readily have this information.

2) Second paragraph of the introduction, it is stated that the brain should become more entropic under the influence of psychedelics and that higher order cortical regions should be impacted to a larger degree than lower level regions. In one of the many reanalyses (<https://doi.org/10.1002/hbm.23234>), it was found that sensory networks exhibited larger increases in entropy than the default mode network (DMN), the network proposed to be at the top of the brain's hierarchy in REBUS (despite other "higher" non-sensory and "lower" sensory level networks exhibiting larger effects under psychedelics).

3) Third paragraph of the introduction and throughout the manuscript, why is there no mention of Preller et al.'s work that has also looked at 5-HT_{2A} expression (<https://doi.org/10.7554/eLife.35082>)? Preller et al. even looked at brain dynamics in relation to 5-HT_{2A} expression over a period of an hour (<https://doi.org/10.1016/j.biopsych.2019.12.027>).

4) Fourth paragraph of the introduction, the authors' psilocybin studies are cited, yet this dataset is not included here.

5) Fifth paragraph of the introduction makes it sound as if the DTI and PET data were from the same participants as the fMRI data, when each of these datasets were on separate samples.

6) First paragraph of the results, the first sentence needs to say that this was a reanalysis, again to minimize the illusion that these 15 participants are a sample independent of prior work.

7) I think it would be worth running k means clustering separately on placebo and LSD data to see whether the same brain states even show up across both conditions or whether the brain states pulled out from this analysis across all data are simply an artifact from the amalgamation of dissimilar data.

8) Figure 2, acronyms should be defined, and what are the units of the radial plots? BOLD activation?

9) First paragraph of LSD Modulates Brain Dynamics section, "...or how often each state appeared per minute (Figure 3b iii), and (iv) transition probabilities..." There is no section (iv) of Figure 3b. If I am not mistaken, all of the analyses of transition probabilities are limited to Figure 4.

10) The fact that dwell times increased at least as strongly as they were decreased under LSD and appearances/min do not appear to be even numerically raised under LSD is not consistent with the findings and interpretations that psychedelics facilitate cycling between brain states. This needs to be discussed.

11) Second paragraph of LSD Modulates Brain Dynamics section, it is a bit odd that the authors refer to the salience network as a bottom-up network. Although it is true that it can respond to salient sensory information, it also supports higher level cognitive functions such as cognitive flexibility. In fact, in the original entropic brain paper, the only changes found were in "the high-level association networks", with the largest effects being in the salience

network (<https://doi.org/10.3389/fnhum.2014.00020>). Another inconsistency within this manuscript itself is that although the authors label meta-state 1 as sensorimotor, it also contains the DMN, the network that should be the most turned off according to REBUS. Moreover, referring to meta-state 2 as frontoparietal is also not accurate as it largely contains bottom-up, lower level, sensory areas (i.e., visual and sensorimotor). This paragraph is overall inaccurate with selective reporting that is consistent with REBUS but failures to mention just as many inconsistencies with REBUS (this is rather true of the REBUS review itself).

12) Regarding the use of other serotonin maps, with recent findings showing the importance of 5-HT1B (<https://doi.org/10.1073/pnas.2022489118>), it is worth mentioning that 5-HT1B was the receptor associated with numerically the next lowest energy, and this did not appear to be significantly different from 5-HT2A.

13) First paragraph of the discussion, as the authors have published, ego dissolution has been shown to correlate with several brain metrics that might or might not be consistent with REBUS and that any single brain-ego dissolution correlation has yet to be replicated. Again, the claim that greater bottom-up activity was found cannot be made.

14) Second paragraph of the discussion, "Since our analysis was carried out on resting state data, it is not surprising that the DMN was prominent across all four brain-states." But this is a bit surprising considering the DMN is supposed to be "turned off" under psychedelics according the REBUS. Perhaps this is more reason to rerun the analyses during music listening.

15) Fourth paragraph of the discussion, "The Entropic Brain Hypothesis...viewing the brain and mind as two sides of the same coin." Not only am I not a fan of such fluffy writing that has become rampant in psychedelic publications, to act like this is some special feature of EBH is to ignore literally all of cognitive neuroscience. Nobody in cognitive neuroscience assumes dualism (at least explicitly).

16) First paragraph of Limitations and Future Work, although it is great that the authors are finally recognizing that they have overanalyzed one small dataset, they use this as an opportunity for self-gratuitous citations that were previously not cited. They then say it should be replicated with different datasets, and the irony is that they have two other psychedelic datasets.

17) Regarding the quantification of energy (i.e., the magnitude of the input that needs to be injected into the brain's structural connectome in order to obtain the desire state transition), does this mean that a given state transition (e.g., state 1 to state 2) is estimated to be the same for every timepoint? This should be clarified.

18) Third paragraph of the Limitations and Future Work, one reason the subjective measures may not correlate is that they occur after scanning. Psychedelics are known to impair recollection (<https://doi.org/10.1007/s00213-018-4981-x>), so such measures may not necessarily be accurate. Another reason is that posthoc subjective measures unreliably correlate with brain measures, and this has been especially true of psychedelic studies (<https://doi.org/10.3389/fpsyg.2018.01475>).

19) It is well-known that pharmacological manipulations and especially psychedelics induce motion artifacts. Such motion artifacts are not typically corrected for with basic motion

regression. For this reason, others have used global signal regression (<https://doi.org/10.1016/j.biopsycho.2019.12.027>) or 24 motion regressors (<https://doi.org/10.1038/s41598-020-73216-8>). Even in non-drug studies and non-clinical populations, it has become default to use global signal regression and/or 24 motion regressors (e.g., <https://doi.org/10.1016/j.neuroimage.2018.11.057>). By default, fMRIPrep produces output for 24 motion regressors. At this point, the use of 6 motion regressors with no other rigorous motion correction method for pharmaco-fMRI data is inappropriate.

20) In the Materials and Methods, “Lastly, time series for 462 gray matter regions were extracted.” Perhaps I’m wrong, but isn’t it 463 regions? In the section Structural Connectivity Network Construction, it is referred to as the “Lausanne-463.”

21) Second paragraph of Characterization of Brain States and Their Hierarchy, by “...extract group-average centroids by taking the mean of all TRs...” Is this the mean of the BOLD signal? If so, is that even comparable across different scans? The arbitrary units of the BOLD signal due to differences in the magnetic field during separate scans is the reason that one cannot compare brain activation patterns of two separate resting state scans.

Reviewer #1 (Remarks to the Author):

LSD flattens the brain's energy landscape: evidence from receptor-informed network control theory

In this article the authors use network control theory to explore the effects of psychedelics on explorations of state space of brain activity. The principal finding is that the space is “flattened”, i.e. there exists a reduction in the energy needed to move from one state to another. This topic is timely and the results compelling. However, there are many methodological concerns that need to be addressed through additional analyses. In general, I have concerns about the robustness of results with respect to reasonable variation in analysis and processing pipelines.

Comments:

1. The authors assume that a single dynamical system describes the evolution of brain activity irrespective of whether the brain is in a psychedelic state or not. Under these dynamics, the energy appears reduced in a psychedelic state. But maybe the actual effect is a change to the dynamics themselves and therefore the optimal trajectories (and their energies) during psychedelic states are incorrect. In fact, the authors own results seem to support this alternative hypothesis – e.g. the increased entropy of brain dynamics.

We thank the Reviewer for their constructive feedback, which has helped us to improve the clarity and rigour of our paper. We hope that the revisions we detail below will satisfactorily address their concerns. First, in response to minor comment #1, we show that our energy landscape findings are insensitive to the choice of k . In our response to minor comment #2, we show that the energy findings are replicated with another clustering algorithm. For #3, we repeat our findings after applying global signal regression (GSR) to the fMRI data, and in response to minor comment #4, we show that if we cluster the two conditions separately, or if we cluster each individual separately, that the energy findings still hold. The latter results indicate that even if the dynamics themselves are changing (i.e. centroids shifting) in the LSD case, the fact remains that the energy required to transition between those shifted states is decreased relative to PL.

Minor comments:

1. A few concerns about the extraction of brain states. The authors justify the solution of $k = 4$ and show that their main findings are replicate with $k = 5$. Do results hold for other k values or just these two? Simply showing the solutions (states) is not sufficient. To make the claim that LSD flattens an energy landscape, the authors need to show that their results are more general and not sensitive on parameterization.

Energy calculations are already shown for $k=5$ in the SI (SI Figure 18c, ii). We have now gone further to show the results for $k=6-8$ (SI Figure 14, a-c). Further, we show that for every choice of k between 2 and 14, the energy landscape - quantified in each individual as the average energy required to transition between all pairs of states - is significantly lowered in LSD (red) compared to PL (black) (SI Figure 14, d).

SI Figure 14: (a-c) Full transition energy comparisons for $k=6, 7,$ and $8,$ respectively. (d) Average placebo (black) and LSD (red) transition energies for each subject, for $k=2:14.$ (Filled, condition mean; ** significant after correction for multiple comparisons).

2. Another concern with k-means clustering. Like many other algorithms, k-means will *always* find a k-partition of a dataset, even if no true clusters exist. Can the authors demonstrate that the states they report are, in fact, separated from one another and that the algorithm hasn't just partitioned into clusters what is essentially a continuous space of activity patterns? At the very least, the authors should test other clustering algorithms to verify that their results are not dependent on the k-means algorithm.

Below we show silhouette scores for LSD and placebo time points from all subjects. Silhouette scores range from -1 to 1 and are computed for each data point, with 1 indicating that a data point is closer to members of its assigned cluster than to members of the next closest cluster, 0 indicating equidistance between the assigned cluster and the closest cluster, and -1 indicating that the data point is closer to another cluster. (a) Silhouette values for data points from 462 independent, normally distributed channels. (b) Silhouette values for data from an independent phase randomized (IPR) null model applied separately to LSD and placebo BOLD data. This null model preserves regional autocorrelation while eliminating non-stationarities and reducing covariance. (c)

Silhouette values for actual LSD and placebo BOLD data. (d) Distribution of silhouette values across all clusters for a null model preserving autocorrelation (red, μ_{ipr}) and for actual data (blue, μ_{real}). A two-sample t-test confirms that silhouette values are larger in the real data, indicating they are indeed clustered in regional activation space beyond what is expected from a random signal with the same regional autocorrelations.

SI Figure 2: (i) Silhouette values for data points from 462 independent, normally distributed channels. (ii) Silhouette values for data from an independent phase randomized (IPR) null model applied separately to LSD and placebo BOLD data. This null model preserves regional autocorrelation while eliminating non-stationarities and reducing covariance. (iii) Silhouette values for actual LSD and placebo BOLD data. (iv) Distribution of silhouette values across all clusters for a null model preserving autocorrelation (red, μ_{ipr}) and for actual data (blue, μ_{real}).

Finally, we replicated our energy results using hierarchical clustering analysis (HCA) rather than k-means to define our brainstates (SI Figure 12). We show a comparison between the brainstates derived from k-means in the main analysis to those derived via HCA (a) and then demonstrate that the transition energy is lower between the HCA states in LSD compared to placebo (b, c).

SI Figure 12: Brain-states derived from HCA demonstrate a flattened landscape. (a) Pearson correlation values and MSE between the 4 states derived from k-means clustering used in the main analysis, and those derived from HCA. (b) Comparison of LSD and placebo energy landscape using the HCA clusters. (c) Comparison of each subject's mean LSD transition energy to their mean placebo transition energy - quantified in each individual as the average energy required to transition between all pairs of states.

3. Another concern about brain states: how does global signal regression impact these results? From the methods section it seems that global signal was not regressed out. Given the contentious nature of this procedure, the authors should verify that global signal regression doesn't impact their main results.

We would like to clarify our processing pipeline; we did indeed de-mean the regional time series for each scan prior to analysis which is not the same as performing global signal regression but it does remove regional and individual level differences in baseline activity. We have modified the text to emphasize this aspect of our processing. In addition, we performed standard global signal regression on the fMRI data and our main results were indeed replicated. We have updated the SI to include the replication with GSR, and modified the methods section to state:

“Regional time-series were de-meaned prior to analysis”.

SI Figure 10: Replication of the main analysis using fMRI data after global signal regression. *significant before multiple comparisons correction, ** significant after multiple comparisons correction.

4. It might be useful to confirm that the brain state estimates are equally applicable for all subjects. One way to do this is to treat each frame as an instant of a state, compute the distance from the state centroid and the actual pattern of activity, and measure the error (e.g. calculate difference between every region's activity, square these values, and add them up). Then take the sum over the entire time series. Are there systematic errors where some subjects are not well-represented by the clusters? Are there differences between the two conditions (LSD/PL)? This is an important point: if the states are more representative of some subjects and less representative of others, then this propagates to the estimates of control energy. The authors need to confirm that the differences in control energies are not driven by differences in how well the states represent an individual.

In order to ensure that our results were not an artifact of clustering two dissimilar physiological conditions, we ran k-means on PL and LSD data separately. We matched the resulting centroids derived from each condition based on their maximum correlation with each other (SI Figure 15, i), and then recomputed the transition energy matrix. We replicated the main paper's finding that most transition energies are significantly lower under LSD compared to placebo. In addition, we compared the average transition energies (over all transitions) between the two conditions and found LSD had significantly lower energies (SI Figure 15, ii). Further, to confirm that differences in control energies are not driven by differences in how well the states represent an individual, we applied k-means to each individual person separately and compared control energies between their respective states. We again ordered centroids based on their correlation to the original 4 clusters in the main analysis and compared individual transitions (SI Figure 15, iii). Again, our main findings hold - LSD lowered the transition energy in all cases. In addition, comparing the average value of the full matrix for the LSD and PL condition we see that overall transition energies are lowered (SI Figure 15, iv).

SI Figure 15: Centroids and energy results from clustering each condition separately (i-ii) and individuals separately (iii-iv).

5. Are there differences in motion/data quality between PL and LSD conditions? How are these differences dealt with?

Yes, there are differences in motion between the two sets of data - unsurprisingly the LSD data has more motion. We first refer the reviewer to our materials and methods section “Data Collection and Processing” where we describe the exclusion criteria: “One subject was excluded due to anxiety, and 4 due to excessive head motion (> 15% of volumes with mean frame-wise displacement > 0.5), leaving 15 subjects (four women; mean age, 30.5 ± 8.0) for analysis.” We have updated the motion statement to include the following which is more detailed regarding motion: “Principally, motion was measured using frame-wise displacement (FD). The criterion for exclusion was subjects with >15% scrubbed volumes when the scrubbing threshold is $FD = 0.5$. After discarding these subjects we reduced the threshold to $FD = 0.4$. The between-condition difference in mean FD for the 4 subjects that were discarded was 0.323 ± 0.254 and for the 15 subjects that were used in the analysis the difference in mean FD was 0.046 ± 0.032 .”

The motion quality of this data set and the measures taken have been detailed extensively in the supplemental information of the original publication (<https://doi.org/10.1073/pnas.1518377113>), which we now summarize in the SI:

“After discarding four subjects due to head motion, fifteen were left for the analyses. There was still a significant between-condition difference in motion for these remaining

subjects however (mean FD of placebo = 0.074 ± 0.032 , mean FD of LSD = 0.12 ± 0.05 , $p = 0.0002$). RSFC analysis is extremely sensitive to head motion (Power et al., 2012) and therefore special consideration was given to the pre-processing pipeline to account for motion. This section goes into more detail about the pre-processing steps that were performed to reduce artefacts associated with motion as well as other non-neural sources of noise.

De-spiking has been shown to improve motion-correction and create more accurate FD values (Jo et al., 2013) and low-pass filtering at 0.08 Hz has been shown to perform well in removing high frequency motion (Satterthwaite et al., 2013). Six motion regressors were used as covariates in linear regression. It was decided that using more than six (e.g., “Friston 24-parameter motion regression” (Friston et al., 1996)) would be redundant and may impinge on neural signal (Bright and Murphy, 2015) (especially when other rigorous processes such as scrubbing (Power et al., 2012) and local white matter (WM) regression were applied (Jo et al., 2010)). Using anatomical regressors is also a common step to clean noise and ventricles, draining vein (DV) and local WM were used in the pipeline employed in the present analyses. Local WM regression has been suggested to perform better than global WM regression (Jo et al., 2013).”

In the main text, our correlations in Figure 6 were performed using the difference in mean FD between LSD and placebo as a covariate of non-interest. We now go a step further and correlate each subject’s mean FD with their fractional occupancy, dwell time, and appearance rate in each cluster, separately for each condition (SI Figures 20 and 21). We do not find any significant correlations. Additionally, we do not find any significant correlation between a subject’s mean FD and their average transition energy for each condition (SI Figure 22).

SI Figure 20: Mean framewise-displacement (FD) of the LSD condition correlated with fractional occupancy (FO), dwell time (DT), and appearance rate (AR). No significant correlations were found, (p uncorrected)

SI Figure 21: Mean framewise-displacement (FD) of the PL condition correlated with fractional occupancy (FO), dwell time (DT), and appearance rate (AR). No significant correlations were found (p uncorrected).

SI Figure 22: Mean framewise-displacement of the LSD (i) and placebo (ii) condition correlated with the average transition energies of each condition. Neither correlation is significant.

We also replicated our main findings after applying GSR, which has been shown to further minimize motion effects since the global signal has been shown to contain signal related to motion (although we note that it may also contain neuronal signal of relevance (<https://doi.org/10.1097/ALN.0000000000003197>)).

6. It is well known that fMRI data is highly autocorrelated, which might explain self-transitions (state X -> state X). Would results hold if the authors treated every temporally contiguous sequence of state X as a single instance of that state? This effectively means that the probability of self-transitions is zero.

The choice of including/excluding self-transitions only impacts the transition probability results, which are not a main emphasis of this paper. Here, we recalculate the group differences in transition probabilities when self-transitions are disregarded. Even though the one off-diagonal TP that had uncorrected significance in the main findings dropped below significance, there is a strong, positive correlation between the off-diagonal entries in the TPs calculated in the two different ways:

SI Figure 23: Our main results and the replications present the transition probabilities that include the probability of staying in the same state (persistence). Here, we present these results (persist; left column) alongside the results when excluding self-transitions (no persist; middle column), and we correlate the off-diagonal of the results obtained from each version (persist vs no persist; right column). (a) LSD main results, (b) psilocybin main results, (c) music replication, (d) z-scored sch232 replication, (e) GSR replication. *significant before multiple comparisons correction, ** significant after multiple comparisons correction, correlation p-values are uncorrected.

7. The authors permute the spatial positions of weights for the 5-HT_{2a} distribution. This is not appropriate, because there exist spatial autocorrelations in the data that are not preserved in the permuted data. The authors should retest this effect using “spin tests” instead (see Markello & Misic 2021).

We appreciate this comment pointing us to the consideration of a more stringent null model. We have now updated the manuscript to show that our results also pass this stronger test (implemented through the Vasa method spin test (10.1093/cercor/bhx249)). This gives us additional confidence in the robustness of our results, and enables us to go beyond what had been shown in previous literature, for which we thank the Reviewer. All of our 2a distribution tests are now compared against this more stringent null model.

8. Weighting structural connections based on streamline counts is not appropriate. All things being equal, larger parcels will tend to be associated with a greater number of streamlines. The authors need to perform some kind of correction in which streamline counts are normalized by some measure of parcel size, e.g. surface area or volume.

We thank the reviewer for this comment. Near-uniform region volume was one reason for the choice of the Lausanne atlas for our main results (10.1016/J.JNEUMETH.2011.09.031). Still, we have now normalized our streamline counts by the sum of voxels contained in each pair of regions. Although this is nearly a constant value in the case of the Lausanne atlas, this is more appropriate to do even if it does not change our results significantly. Energy calculations have been corrected accordingly.

References:

Markello, R. D., & Misic, B. (2021). Comparing spatial null models for brain maps. *NeuroImage*, 236, 118052.

Reviewer #3 (Remarks to the Author):

In another reanalysis alongside the 15+ other reanalyses of this dataset (more publications than participants), Singleton and colleagues provide partial evidence for the REBUS model. Although this may be one of the better tests of REBUS, the salami slicing has gotten out of control. Not only is the small sample (N = 15) not independent of all the other tests of REBUS, the analyses themselves are not necessarily independent (e.g., <https://doi.org/10.1016/j.neuroimage.2020.117049> in which similar complexity measures were applied to these data). Moreover, whereas some of this group's past work has occasionally included their psilocybin dataset (e.g., both included in a single publication <https://doi.org/10.1016/j.neuroimage.2020.117049> and <https://doi.org/10.1016/j.cub.2016.02.010>; salami sliced into two separate publications <https://doi.org/10.1038/s41598-017-17546-0> and <https://doi.org/10.1016/bs.pbr.2018.08.009>; see also <https://doi.org/10.1016/j.neuroimage.2019.05.060> for analysis of only the psilocybin dataset when multiple publications of the LSD dataset had already been out), the same analyses to the psilocybin dataset are conspicuously missing (as they tend to be from most of the reanalyses). If I am not mistaken, this group now also has a DMT dataset that together could essentially triple their sample size and provide some generalizability to these findings. REBUS is supposed to be an all-inclusive model of psychedelic (i.e., 5-HT_{2A}) drug action rather than specific to LSD.

Additionally, if what is found here is thought to be a general principle of psychedelic drug action on the brain, then the listening to music condition should also be expected to exhibit such effects. Although this paper should not be published in *Nature Communications* (or anywhere until all other datasets are integrated), wherever it may end up, the following points must be considered:

Of course, as a classified substance in most countries (including the UK, where this dataset was collected) it is very difficult and expensive to collect such data. This, together with ethical considerations, also explains the limited sample size, which is well in the range of other published datasets on psychedelic drugs. Despite this limited sample size, our results are not only statistically robust but also (now, in this response) replicated under several different conditions including using global signal regression, task music scans, different clustering choices and an independent dataset (specifically, the much referenced psilocybin data in the comment above). All in all, it seems that, no matter how you slice it, energy is reduced under LSD and psilocybin in support of the REBUS hypothesis.

We note that to the best of our knowledge, the LSD fMRI dataset we use here is the largest one that has been made publicly available, which explains its wider use. We find it

very bizarre that the Reviewer should penalise us for the original PI's good scientific practices of data-sharing. We also point out that other public datasets such as the Human Connectome Project have been analysed extensively, with over 3000 publications and a sample size of only $N = 1200$ individuals (which certainly qualifies the HCP data as having more publications than subjects). We do not believe that this means all future work using the HCP data is null and void. With regards to the DMT fMRI dataset - this dataset is not published anywhere yet, and it will be the subject of a dedicated publication in the future. Perhaps once the PIs release it - adhering to their rare, exceptional dedication to open science - a replication study using our methods applied to their data would be an appropriate future study.

We do not consider it sound scientific practice to forbid publication of re-analyses of existing data - even more so with such rare datasets that are extremely difficult to acquire anew. Analytic methods evolve and improve, making it possible to view the same data in a new light, ask new questions that could not have been addressed before, and ultimately extract new knowledge that had not previously been obtained from the data. We believe that so long as new knowledge can be obtained from a dataset, that dataset should be re-analysed. The Reviewer agrees that "this may be one of the better tests of REBUS", and this was precisely our aim. This being said, we agree of course that replication would strengthen our results, and we now have done that in several different ways, including analysis of a new dataset (the aforementioned psilocybin data), analysis of the additional task fMRI music-listening scans (in the same LSD subjects), and several robustness replications in the LSD dataset using different processing choices and clustering choices (see SI and new main text Figures 3, 4, and 5).

We have updated the SI to include the cluster centroids and energy results from a separate analysis of the music scans which show a flattened energy landscape under LSD:

SI Figure 7: A within-subject replication of the main results using fMRI scans taken while the individuals listened to music. We find similar brain-states to the main resting-state analysis and a flattened energy landscape under LSD.

1) The abstract needs to state that this is another reanalysis. This point is generally skipped across all of the reanalyses, giving the illusion of independent samples. A trainee recently questioned whether all of Imperial's fMRI studies were actually based on only two datasets of 15 subjects, and although such information is typically found in the methods, many people simply read abstracts and do not have ties to the researchers themselves to readily have this information.

Once again, this comment about “all of the reanalyses” is clearly a criticism of some of the co-authors’ publishing practices, and not about the current paper’s merits or limitations. We always welcome open, constructive discussion on how to improve scientific practices - in the appropriate venue.

Information about this specific paper being a reanalysis is clearly provided in our manuscript and regularly in other papers using this data; we have added mention of this fact in the abstract as well to adhere to the utmost standard of transparency for the superficial reader.

2) Second paragraph of the introduction, it is stated that the brain should become more entropic under the influence of psychedelics and that higher order cortical regions should be impacted to a larger degree than lower level regions. In one of the many reanalyses (<https://doi.org/10.1002/hbm.23234>), it was found that sensory networks exhibited larger increases in entropy than the default mode network (DMN), the network proposed to be at the top of the brain’s hierarchy in REBUS (despite other “higher” non-sensory and “lower” sensory level networks exhibiting larger effects under psychedelics).

From the paragraph which the reviewer is referring to: “The REBUS model postulates that psychedelics alter cognitive functioning by serotonergic action at 5-HT_{2a} receptors in higher-order cortical regions. Dysregulation of these regions’ activity results in a weaker effect of prior beliefs and expectations in shaping the interpretation of bottom-up information, ultimately allowing the brain to explore its dynamic landscape more readily - as suggested by more diverse (entropic) brain activity.”

Nowhere do we state that the entropy of higher cortical regions should be impacted more than those of lower level regions. We merely point to psychedelic action at high-level regions as the likely source for the larger effects that may also involve lower-order sensory networks. As we state, the hypothesis put forth by REBUS is that disruption at the highest level leads to global dysregulation. The reviewer here appears to have misinterpreted our meaning ; we have added clarification to the text that emphasizes our point.

“The REBUS model postulates that psychedelics alter conscious experience via their agonist action at 5-HT_{2a} receptors - which have especially high expression in higher-order cortical regions. Agonist-induced dysregulation of spontaneous activity in these regions is postulated to translate into decreased precision-weighting on prior beliefs - which has reciprocal (enabling) implications for bottom-up information flow.”

3) Third paragraph of the introduction and throughout the manuscript, why is there no mention of Preller et al.’s work that has also looked at 5-HT_{2A} expression (<https://doi.org/10.7554/eLife.35082>)? Preller et al. even looked at brain dynamics in relation to 5-HT_{2A} expression over a period of an hour (<https://doi.org/10.1016/j.biopsych.2019.12.027>).

We would kindly like to note that we actually did cite three of Preller et al's papers on 5-HT2a expression during the results paragraph on 2a, including the eLife article the reviewer provided.

“This is especially noteworthy because serotonin 2a receptor agonism plays a prominent role in how LSD and other classic psychedelics influence neural activity and subjective experience^{6–8,43–47}.”

We must note that the third paragraph of the introduction is commenting explicitly on computational models relating dynamic brain activity to 2a expression (the subject matter of the current analysis).

4) Fourth paragraph of the introduction, the authors' psilocybin studies are cited, yet this dataset is not included here.

We have now replicated our main findings using this independent dataset, see new Figures 3, 4, and 5, results/discussion and extensive SI changes.

5) Fifth paragraph of the introduction makes it sound as if the DTI and PET data were from the same participants as the fMRI data, when each of these datasets were on separate samples.

We have changed the phrasing accordingly to include the word “separate”, see below. We note that all the relevant information is provided in the Methods.

“...we combine functional MRI data from two datasets comparing drug (LSD or psilocybin) to placebo, with separate structural (white-matter) connectivity networks obtained from diffusion MRI (dMRI), and receptor density maps from positron emission tomography (PET)”

6) First paragraph of the results, the first sentence needs to say that this was a reanalysis, again to minimize the illusion that these 15 participants are a sample independent of prior work.

We have changed the phrasing accordingly. Again, we note that all the relevant information is provided in the Methods, which we deem to be an essential part of the paper.

7) I think it would be worth running k means clustering separately on placebo and LSD data to see whether the same brain states even show up across both conditions or whether the brain states pulled out from this analysis across all data are simply an artifact from the amalgamation of dissimilar data.

Please see our response to R1's minor comment #4. We show that even when clustered separately, the LSD trajectories have lower energy than PL. Additionally, we show reduced energy when clustering individuals separately, when using a different clustering method and when using between 2-14 clusters. We believe that these results support the robustness of our main findings.

8) Figure 2, acronyms should be defined, and what are the units of the radial plots? BOLD activation?

The acronyms for the resting state networks are now defined in the caption. We have also clarified the caption to more clearly indicate that the discussed cosine similarity measure is referring to the radial plots:

“Group-average recurrent brain-states are represented by the mean activation pattern across all subjects and conditions for each of the 4 clusters (brain representations at the bottom of the figure). For each brain-state, we separately calculated the cosine similarity (radial plots) of its high-amplitude (supra-mean) activity and low-amplitude (sub-mean) activity to a priori resting-state networks (RSNs); resulting similarity measures are represented via radial plots²⁹. Meta-State 1 (MS-1) is composed of two sub-states (MS-1a and MS-1b) which are characterized by the contraposition of the somatomotor and ventral attention/salience networks with the default-mode network, whereas the Meta-State 2 (MS-2; MS-2a and MS-2b) is characterized by the contraposition of the default-mode, somatomotor and visual networks with the frontoparietal network. The dichotomy of these states can be observed visually in the radial plots and on the rendered brain volumes, and is confirmed via their negative, significant Pearson correlation (SI Figure 3, ii). States shown here are from the LSD dataset. States derived from the psilocybin dataset are highly similar (SI Figure 4). DAT dorsal attention network, DMN default mode network, FPN frontoparietal network, LIM limbic network, SOM somatomotor network, VAT ventral attention network, and VIS visual network.”

9) First paragraph of LSD Modulates Brain Dynamics section, “...or how often each state appeared per minute (Figure 3b iii), and (iv) transition probabilities...” There is no section (iv) of Figure 3b. If I am not mistaken, all of the analyses of transition probabilities are limited to Figure 4.

We have modified the sentence in question to be more clear to the reader. Please note some labeling has changed to incorporate the psilocybin results to the main paper.

“From each subject’s temporal sequence of brain-states (Figure 3a) we obtained a systematic characterization of the temporal dynamics of the 4 states, namely, their fractional occupancies, or the probability of occurrence of each state (Figure 3b/c, i), dwell times, or the mean duration that a given state was maintained, in seconds (Figure 3b/c, ii), appearance rates, or how often each state appeared per minute (Figure 3b/c, iii), and transition probabilities, or the probability of switching from each state to every other state (Figure 4a/b, i). ”

10) The fact that dwell times increased at least as strongly as they were decreased under LSD and appearances/min do not appear to be even numerically raised under LSD is not consistent with the findings and interpretations that psychedelics facilitate cycling between brain states. This needs to be discussed.

This has already been discussed. Individual-level changes in overall dwell time and appearance rates correlated with energy reduction by LSD, and now, we additionally show trends with psilocybin as well. In light of the psilocybin results, we have updated the main text: “Our results show that the more LSD lowered the average transition energy of a given subject, the more the empirically observed dwell times decreased and the more the empirically observed appearance rates increased. This is particularly interesting, as there were no group-level differences in appearance rates between the two conditions, and a mix of increased and decreased dwell times depending on the state. While the correlations identified are consistent with our hypothesis of a flattened energy landscape, where lower barriers between brain-states results in increased frequency of state transitions and shorter state dwell times, it seems this individual-level effect does not translate to a group difference in dwell time or appearance rate (SI Figure 5a). Interestingly, the psilocybin data had the same directionality of correlation between energy reductions and decreased dwell time ($r = 0.444$, $p = 0.271$; $n = 9$) and increased appearance rate ($r = -0.453$, $p = 0.259$, $n = 9$) (SI Figure 4c). However, these data had the additional characteristic of group-level reductions in dwell time and increases in appearance rate under psilocybin compared to placebo that are more straightforward to interpret under the REBUS hypothesis (SI Figure 5b). Future work with a larger number of subjects may allow further interrogation of this phenomena.”

11) Second paragraph of LSD Modulates Brain Dynamics section, it is a bit odd that the authors refer to the salience network as a bottom-up network. Although it is true that it can respond to salient sensory information, it also supports higher level cognitive functions such as cognitive flexibility. In fact, in the original entropic brain paper, the only changes found were in “the high-level association networks”, with the largest effects being in the salience network (<https://doi.org/10.3389/fnhum.2014.00020>). Another inconsistency within this manuscript itself is that although the authors label meta-state 1 as sensorimotor, it also contains the DMN, the network that should be the most turned off according to REBUS. Moreover, referring to meta-state 2 as frontoparietal is also not accurate as it largely contains bottom-up, lower level, sensory areas (i.e., visual and sensorimotor). This paragraph is overall inaccurate with selective reporting that is consistent with REBUS but failures to mention just as many inconsistencies with REBUS (this is rather true of the REBUS review itself).

We believe that the reviewer must be taking issue with the title of this section (“LSD modulates brain dynamics by increasing occupancy in bottom-up,

somatomotor-dominated brain-states”) since we do not refer to the states as bottom-up within the referenced paragraph. To address this concern, we have changed the title to “*LSD modulates brain dynamics by increasing occupancy in MS-1 and decreasing occupancy in MS-2*” (see below for new naming convention).

Regarding the criticism on labeling, we note that our states were labeled as per the data-driven method adopted by Cornblath 2020, such that the labels reflect the prevalence of each RSN in the data, rather than our own interpretation. We also note in the text that the labeling is used only for convenience, and did not guide the analysis. We also show the full states in Figure 2, as both radial plots and brain projections, for anyone to reference.

However, due to the unsettled debate on the interpretation of these co-activation patterns, we have walked back on our claims of finding increased bottom-up activity in the abstract and discussion and have relabeled our states to be more agnostic with the dual purpose of easing the reader through results obtained from different datasets and processing choices. The original 4 centroids in Figure 2 are now labeled as MS-1a and MS-1b (formerly SOM- and SOM+) for the first meta-state, and MS-2a and MS-2b (formerly FPN- and FPN+). In all re-analyses (either with the psilocybin data or with different processing/parameter choices in the SI), we have ordered and labeled the states based on their maximum correlation to these original 4 states.

The paragraph in question now reads as follows:

“We found that for both psychedelic and placebo conditions, the brain most frequently occupies MS-1 (higher fractional occupancy) whose constituent sub-states are also visited for the longest periods of time (highest dwell times) (Figure 3b). LSD modifies the fractional occupancy of these states by decreasing the dwell times of MS-2 and further increasing dwell times of the already dominant MS-1 (Figure 3b).”

Regarding the reviewer’s comment that the DMN should be “turned off”, we note that nowhere is it proposed or shown in the literature that the DMN becomes inactive, rather, it becomes less functionally connected (<https://www.pnas.org/content/113/17/4853>) and has nodal (voxel-level) decreases in activity (<https://www.pnas.org/content/109/6/2138>). We point to our response on this topic in Reviewer 3’s comment #14. Referring to networks as “turned off” is misleading and a misunderstanding of the relevant dynamic changes, and it is not what the REBUS hypothesis states.

12) Regarding the use of other serotonin maps, with recent findings showing the importance of 5-HT1B (<https://doi.org/10.1073/pnas.2022489118>), it is worth mentioning that 5-HT1B was the receptor associated with numerically the next lowest energy, and this did not appear to be significantly different from 5-HT2A.

We thank the reviewer for this comment and have added the following to the results section:

“Notably, the 5-HT1b receptor resulted in the second largest energy reduction in our model. This is of particular interest as recent animal models have implicated this receptor as the the potential site of antidepressant action of selective serotonin reuptake inhibitors (SSRIs)⁴⁹ as well as a potential route through which serotonergic psychedelics like psilocybin enact their synaptogenesis and antidepressant effects^{50,51”}

However, we would like to clarify that the 2a energies were lower than the 1b energies (and every other receptor tested) as indicated by the asterisks at the very top of Figure 5 and explicitly mentioned in the main text and figure caption. We assume that the reviewer was confused by the representation of outliers as dots outside of the boxplots, and misinterpreted them as indicating significance. We have reduced the size of these outlier points, now indicate in the caption that the dots are outliers and further emphasize the significant differences with red double asterisks and line at the top. In addition, this panel has now been moved to its own figure to show alongside the psilocybin results.

Figure 5: The serotonin 2a receptor flattens the energy landscape more than any other receptor tested. We weighted our model with expression maps of other serotonin receptors (5-HT1a, 5-HT1b, and 5-HT4), and the serotonin transporter (5-HTT), and found that 5-HT2a resulted in significantly lower transition energy (averaged across all pairs of states) than all others. (i) Energies calculated from each subject’s placebo centroids in the LSD dataset (n = 15). (ii) Energies calculated from each subject’s placebo centroids in the psilocybin dataset (n = 9). • outlier, ** significant after multiple comparisons correction.

13) First paragraph of the discussion, as the authors have published, ego dissolution has been shown to correlate with several brain metrics that might or might not be consistent with REBUS and that any single brain-ego dissolution correlation has yet to be replicated. Again, the claim that greater bottom-up activity was found cannot be made.

In the paragraph in question, we state the following about ego dissolution: “This is theorized to correspond to a flattening of the functional hierarchy as well, i.e. a relaxation of the weighting that high-level priors exert on inputs from lower-level (sensory) regions - thought to be a pivotal component of psychedelics’ therapeutic mechanism of action as well as characteristic subjective effects of ego dissolution and visual/auditory distortions.”

We have now simplified and revised the relevant paragraph pertaining to ego-dissolution:

“This is theorized to correspond to a flattening of the functional hierarchy as well, i.e. a relaxation of the weighting of high-level priors - thought to be a pivotal component of psychedelics’ therapeutic mechanism.”

The reviewer should note that we have removed our claims that greater bottom-up activity was found.

14) Second paragraph of the discussion, “Since our analysis was carried out on resting state data, it is not surprising that the DMN was prominent across all four brain-states.” But this is a bit surprising considering the DMN is supposed to be “turned off” under psychedelics according to the REBUS. Perhaps this is more reason to rerun the analyses during music listening.

DMN “disintegration” is a functional connectivity measure - specifically a measure of decreased *within-network* connectivity. We would never use “turned off” as a descriptor for this change in network dynamics. The fact that DMN regions are less functionally connected to each other under psychedelics does not mean that they are not active; it means their activity patterns become decoupled from each other. Additionally, if the reviewer is referring to the reported voxel-wise spatial pattern showing BOLD decreases in some DMN nodes after psilocybin administration [<https://www.pnas.org/content/109/6/2138>], they should note that our regional analysis is unlikely to have a 1 to 1 correspondence with voxel-level results. We have also now validated our results separately with the music scans (SI Figure 7, provided in the main response to Reviewer 3).

We have added the following to the discussion and removed our claims of increased bottom-up activity:

“It should be noted that this regional-level, activation-based, study is not expected to capture the previously reported DMN disintegration³³ (a functional connectivity measure), nor the reduced voxel-level activity of some DMN nodes³⁴ under psychedelics.”

15) Fourth paragraph of the discussion, “The Entropic Brain Hypothesis...viewing the brain and mind as two sides of the same coin.” Not only am I not a fan of such fluffy writing that has become rampant in psychedelic publications, to act like this is some special feature of EBH is to ignore literally all of cognitive neuroscience. Nobody in cognitive neuroscience assumes dualism (at least explicitly).

The phrase that we use is specifically referring to the paper by Northoff et al (2020) in *Physics of Life Reviews* (cited in that very same sentence), which in its very title addresses the question “Is temporo-spatial dynamics the “common currency” of brain and mind?” - clearly a topic of extreme relevance to our work, and the source of our coin/currency metaphor.

This is a recent paper that has generated very lively discussion among several researchers in cognitive neuroscience, physics and beyond (see 9 commentaries in the same issue of *Physics of Life Reviews*). Therefore, we strongly reject the claim that we are “ignoring all of cognitive neuroscience”, for which there is simply no ground: on the contrary, we are engaging with some of the most recent debates in the rich and fast-moving literature at the intersection of cognitive neuroscience and other disciplines.

16) First paragraph of Limitations and Future Work, although it is great that the authors are finally recognizing that they have overanalyzed one small dataset, they use this as an opportunity for self-gratuitous citations that were previously not cited. They then say it should be replicated with different datasets, and the irony is that they have two other psychedelic datasets.

We cite these works because we consider them relevant, and the Reviewer does not seem to disagree on their relevance - only the fact that they used the same dataset. We also disagree with the Reviewer’s implication that we “recognise that we have overanalysed one small dataset”: in the manuscript we do not talk about “overanalysis”, and as we have argued above, we fundamentally disagree with this characterisation, because the analyses presented here are new and so are the insights we obtained from them - even as we recognise the importance of acquiring and testing new data.

As for the replication, we now provide it with psilocybin data - the only other dataset that is currently available to us.

17) Regarding the quantification of energy (i.e., the magnitude of the input that needs to be injected into the brain’s structural connectome in order to obtain the desire state transition), does this mean that a given state transition (e.g., state 1 to state 2) is estimated to be the same for every timepoint? This should be clarified.

We have added clarification both to the section in which the Reviewer is referring to, and to the methods section:

Results: “For each subject and condition, we calculated the energy needed to transition between each pair of brain-states. On an individual level, brain-states were defined as the centroid of all TRs assigned to each state during that person’s LSD or placebo scans.”

Methods: “We can extract 1) group-average centroids by taking the mean of all TR’s assigned to each cluster (all subjects, all conditions) (shown in Figure 2), 2) condition-average centroids by taking the mean of all TR’s assigned to each cluster separately for each condition (shown in SI Figure 6; placebo condition-average centroids were used for Figure 4a, iv and Figure 4b calculations), and 3) individual condition-specific centroids by taking the mean of all TRs assigned to each cluster for a single subject and condition (used for Figure 4a, ii & iii calculations).”

18) Third paragraph of the Limitations and Future Work, one reason the subjective measures may not correlate is that they occur after scanning. Psychedelics are known to impair recollection (<https://doi.org/10.1007/s00213-018-4981-x>), so such measures may not necessarily be accurate. Another reason is that posthoc subjective measures unreliably correlate with brain measures, and this has been especially true of psychedelic studies (<https://doi.org/10.3389/fpsyg.2018.01475>).

We thank the Reviewer for these helpful suggestions, which we have included in our discussion:

“Psychedelics have been found to impair some aspects of memory recollection in humans⁷⁰ (though not autobiographical memory recollection⁷¹), which may arguably impact on the fidelity of post-hoc subjective ratings and how they correlate with acute, objective brain-measures.”

19) It is well-known that pharmacological manipulations and especially psychedelics induce motion artifacts. Such motion artifacts are not typically corrected for with basic motion regression. For this reason, others have used global signal regression (<https://doi.org/10.1016/j.biopsycho.2019.12.027>) or 24 motion regressors (<https://doi.org/10.1038/s41598-020-73216-8>). Even in non-drug studies and non-clinical populations, it has become default to use global signal regression and/or 24 motion regressors (e.g., <https://doi.org/10.1016/j.neuroimage.2018.11.057>). By default, fMRIPrep produces output for 24 motion regressors. At this point, the use of 6 motion regressors with no other rigorous motion correction method for pharmaco-fMRI data is inappropriate.

The Reviewer appears to incorrectly state that we used “6 motion regressors with no other rigorous motion correction”. Our Methods clearly report that we also included 3 anatomical regressors. The anatomical regressors we used take into account physiological and motion noise. We have clarified this further in the Methods and expanded on their relevance, although we note that extensive information in this respect is present in the original publication, to which we refer in the text. We also further addressed motion by including scrubbing and rejecting high-motion subjects; this

information is also provided in the manuscript. Finally, in response to R1, we included motion as a covariate of no interest in our analyses. Please see the full response to R1 for more on motion quality and the efforts taken to address it. In particular, we now show that the results remain unchanged if GSR is used - although we must disagree with the claim that “it has become default to use global signal regression”, as this is still far from a settled debate, with recent reports that the global signal may in fact contain neuronal signal of relevance for consciousness (<https://doi.org/10.1097/ALN.0000000000003197>).

20) In the Materials and Methods, “Lastly, time series for 462 gray matter regions were extracted.” Perhaps I’m wrong, but isn’t it 463 regions? In the section Structural Connectivity Network Construction, it is referred to as the “Lausanne-463.”

The 463rd region (brain stem) was not included in the analysis, as reported in the original publication by Cornblath et al (2020), whose procedures we followed. This information is reported in the same sentence that the reviewer is referring to: “Lastly, time series for 462 gray matter regions were extracted (Lausanne scale 4, sans brain-stem).” We have now added clarification to the SC Network Construction section to avoid confusion.

21) Second paragraph of Characterization of Brain States and Their Hierarchy, by “...extract group-average centroids by taking the mean of all TRs...” Is this the mean of the BOLD signal? If so, is that even comparable across different scans? The arbitrary units of the BOLD signal due to differences in the magnetic field during separate scans is the reason that one cannot compare brain activation patterns of two separate resting state scans.

As we report in the first paragraph of the section in question, the data was demeaned prior to analysis such that for each scan, a region’s mean over time is 0 - as per the paper by Cornblath et al (2020). In addition, we have included results for z-scored data, and data with GSR in the SI. We hope that this clarification and the additional analyses will satisfy the Reviewer that our methodology and the resulting discoveries are sound.

Reviewers' comments:

Reviewer #1 (Remarks to the Author):

In their response, the authors addressed most of my questions. However, there are a few where my comment was either misunderstood or not fully addressed.

I also agree with the authors that Reviewer #3 is being needlessly antagonistic concerning the reuse of existing data. Reusing data is common practice in imaging studies (and really every scientific domain, especially where data is costly or difficult to collect). It's especially common in studies of hard-to-sample populations or where the experimental design requires special care, e.g. ingesting controlled substances.

--

In my comment #1, I suggested that maybe the dynamics are shifted in the psychedelic state. That is, that some aspect of the $dx/dt = Ax + Bx$ used to describe the temporal evolution of controls is different when taking psychedelics. For instance, maybe connections (the A matrix) are reweighted or their contributions to the dx/dt reduced. If that were the case, then maybe given this new dynamical system the trajectories would appear optimal (or near-optimal), but because the authors use the same equation for control/psychedelic conditions they can't tell. I recognize that addressing this fully is likely impossible. But the authors should still comment.

In my second comment I suggested performing some test to confirm that the clusters are objectively "good". The authors generated null data and showed that the silhouette scores of the null data are smaller than that of the actual data. This does not address my comment because the null data have different similarities to one another than the observed. One possibility would be to select a random point in the real data to treat as a centroid. Then select another that's maximally distant from it. Then another that's maximally distant from both, and so on. Then recalculate silhouette scores. Note that the cluster centroids estimated this way are not optimized and only represent points that are reasonably dissimilar/distant from one another.

In my fifth comment, I raised concerns about motion. Specifically, I was interested in how the authors dealt with motion on a frame-by-frame basis. This is an important point because the authors cluster individual frames to determine "states". It would be useful if the authors censored those high-motion frames prior to clustering to ensure that they have little/no impact on centroid estimation.

Reviewer #4 (Remarks to the Author):

Singleton et al. present a compelling empirical evidence showing that (classical) psychedelics flatten the energy landscape of brain dynamics resulting in an agile transition between multiple brain states. This shows the brain's flexibility to visit (or revisit) various states under psychedelics hence demonstrating evidence for the therapeutic benefits these drugs can provide for people with internalising disorders which are marked by the trapping / rigidity to be stuck in a certain brain state. I didn't review the previous submission but I see

that reviewers provide an extensive feedback which has resulted in a substantially improved paper especially in terms of the replication on two separate datasets one with psilocybin and other with music listening component. In that sense authors have done a very good job.

However, I do have one concern which wasn't raised in the previous round of review. This is regarding the rather amusing twist of the technical terminology. The RHEBUS hypothesis, which this paper is claiming to test, is specifically about the flattening of the "free" energy landscape while the current manuscript dropped "free" and talks about flattening of energy landscape. While this seems innocuous enough to a cursory reader however for a more technically minded readers there is clear discrepancy. Free energy is an information theoretic construct (also referred to as 'surprise' within the FEP framework), formally defined as the lower bound on the (log)model evidence. This reflects a cost function which is optimised when fitting a (generative) model to observed data. Whereas the (transition) energy metric used in the current manuscript is an entirely different construct defined as the "minimum energy required for the brain to transition from one activation pattern to another". The RHEBUS paper (I am sorry that I need to describe it here for the author) combined entropic brain hypothesis and FEP which are both information theoretic arguments. An excerpt from RHEBUS: "The entropic brain hypothesis and free-energy principle are inter-related, not least because of their shared appeal to information theoretical metrics, closely linked to classic (Shannon) entropy:" and "The entropic brain measures the uncertainty of neuronal fluctuations across time, whereas free-energy measures the uncertainty of beliefs encoded by neuronal fluctuations." Current paper claims to test RHEBUS but then goes on to test a different quantity that was never mentioned in that paper which only talked about free energy. Free energy in RHEBUS and energy metric used in this paper to me are chalk and cheese and at best be only colloquially compared perhaps in the Discussion section. Transition energy as a proxy for free energy is a stretch too far. The only way I see that RHEBUS can be tested is using some constructs that appeal to model optimization and comparison (as is done in DCM). I would suggest that authors either find a way to resolve this technical discrepancy (which I am not sure is possible as this would mean going back in time and changing the RHEBUS hypothesis) or don't emphasise that this paper is testing RHEBUS which is more tenable position in my opinion. The presented evidence is very compelling on itself, replicated across datasets and can be guided by a rich literature on analysis of energy landscape as measure by fMRI (dozens of papers show up on the topic without a recourse to free energy optimization). I would suggest rewriting and repositioning of the current manuscript such that claims of testing RHEBUS (which is tied to testing hypothesis based on free energy) are not front and centre (probably only warranted in Discussion where speculative connections can be made).

Reviewer #5 (Remarks to the Author):

None.

REVIEWER COMMENTS

Reviewer #1 (Remarks to the Author):

In their response, the authors addressed most of my questions. However, there are a few where my comment was either misunderstood or not fully addressed.

I also agree with the authors that Reviewer #3 is being needlessly antagonistic concerning the reuse of existing data. Reusing data is common practice in imaging studies (and really every scientific domain, especially where data is costly or difficult to collect). It's especially common in studies of hard-to-sample populations or where the experimental design requires special care, e.g. ingesting controlled substances.

We would like to thank Reviewer 1 for their continued contribution to the careful and thoughtful review of our manuscript. We appreciate their constructive thoughts and feedback. We have addressed the clarifying questions below, and apologize for our initial misunderstanding.

--

In my comment #1, I suggested that maybe the dynamics are shifted in the psychedelic state. That is, that some aspect of the $dx/dt = Ax + Bx$ used to describe the temporal evolution of controls is different when taking psychedelics. For instance, maybe connections (the A matrix) are reweighted or their contributions to the dx/dt reduced. If that were the case, then maybe given this new dynamical system the trajectories would appear optimal (or near-optimal), but because the authors use the same equation for control/psychedelic conditions they can't tell. I recognize that addressing this fully is likely impossible. But the authors should still comment.

We thank the reviewer for this clarification. Here, we assume that both conditions are equally well described by the same equation. We believe that this assumption is justified as the same equation has been used consistently in other network control theory literature for rest and tasks, but also depression and schizophrenia, demonstrating its broad applicability. Nevertheless, we acknowledge that this is an implicit limitation of our analytic framework, and we have added discussion on this in the same paragraph where we discuss the assumption of linearity in the Limitations section:

“Finally, our approach is based on network control theory, which differs from other recent computational investigations using e.g. whole-brain simulation through dynamic mean-field modeling of brain activity^{6–8,23}. These latter approaches employ a neurobiologically realistic model of brain activity based on mean-field reduction of spiking neurons into excitatory and inhibitory populations, and have been used to account for non-linear effects of 5HT_{2a} receptor neuromodulation induced by psychedelics. In contrast, network control theory relies on a simpler linear model, which we employed due to its ability to address our prediction that transition (control) energies

would be reduced under psychedelics. Additionally, recent evidence suggests that most of the fMRI signal may be treated as linear^{75,76}. **By using the same linear equation for modeling both the psychedelic and placebo states, we quantify relative changes in the energy landscape while keeping the dynamics of the model the same for each case; implicitly, this also means that we assume that the underlying structural connectivity is providing the same contribution in both conditions. However, using different models for the two cases may represent an avenue for future research.** Combining both approaches to capitalize on the strengths of each will be a fruitful avenue for future work.”

In my second comment I suggested performing some test to confirm that the clusters are objectively "good". The authors generated null data and showed that the silhouette scores of the null data are smaller than that of the actual data. This does not address my comment because the null data have different similarities to one another than the observed. One possibility would be to select a random point in the real data to treat as a centroid. Then select another that's maximally distant from it. Then another that's maximally distant from both, and so on. Then recalculate silhouette scores. Note that the cluster centroids estimated this way are not optimized and only represent points that are reasonably dissimilar/distant from one another.

Thank you for this suggestion. We take the reviewer's comment as pointing out that, through optimization, *k*-means might provide reproducible decision boundaries (SI Figure 1) even if the data has no obvious (to the eye) cluster-like structure. We have performed the suggested procedure, using the first randomly selected point as the centroid for Cluster 1, the maximally distant point from it as the centroid for Cluster 2, and so on defining centroids for Clusters 3 and 4. We then assign all remaining points to the closest centroid from the prior step. This suggested process, lacking optimization, should tell us something about the data structure.

If, on one extreme, the data were ideal and contained clusters that were optimally separated from each other (with between-cluster distance naturally larger than within-cluster distance), then we would expect the above process to result in a partition with high silhouette scores on the order of those achieved through *k*-means. If, on the other hand, there was no structure to the data at all, we should expect that this process would generate partitions with low silhouette values, on the order of those contained in our null model (SI Figure 2).

We generated 1,000 of these non-optimized partitions using the process above, starting from a different random initial point each time. The average silhouette score of non-optimized partitions was less than that created from *k*-means ($\mu_{\text{opt}} - \mu_{\text{non-opt}} = 0.0379$, $p < 0.001$) and greater than the average silhouette score created from applying *k*-means to the independent phase-randomized (IPR) null model ($\mu_{\text{non-opt}} - \mu_{\text{IPR}} = 0.0468$, $p < 0.001$). This suggests that our real data is (unsurprisingly) not equivalent to a perfectly idealized case such as the one described above. Nevertheless, it does contain identifiable structure, significantly exceeding what is observed in our auto-correlation-preserving null model. Since our data is based on fMRI measurements which reflect smoothly varying

hemodynamic responses, we believe one may expect some measurements to occur at the boundary between states. Even if neuronal populations instantaneously switch between states, the brain activity patterns measured by BOLD will be blurred by the hemodynamic response function, resulting in some measurements occurring at the boundaries between states. Finally, we would like to note that k -means has been applied to brain data several times in the literature (see citations at the end of the insert below); despite also having the issue of non-optimally separated brain activity data, these works have reported behaviorally or physiologically meaningful findings. We have added the following text to the limitations to discuss this issue:

“One other consideration is the objective quality of the clustering - because fMRI measures the smoothly varying hemodynamic response to neuronal activation (BOLD)^{72,73}, there may be suboptimal separation of clusters. For example, even if neuronal activity does switch between states near-instantaneously, there will still be a smoothing of the measured activity in the fMRI which would result in some TRs occurring at the boundaries of the states. Despite this potential limitation, our current work and previous studies on dynamic states from fMRI have revealed physiologically and behaviorally meaningful cluster-based metrics^{8,8,29,30,74-76}.”

In my fifth comment, I raised concerns about motion. Specifically, I was interested in how the authors dealt with motion on a frame-by-frame basis. This is an important point because the authors cluster individual frames to determine "states". It would be useful if the authors censored those high-motion frames prior to clustering to ensure that they have little/no impact on centroid estimation.

We apologize for the lack of clarity in our description of preprocessing. We included "scrubbing" as a step in our methods, while omitting what was meant by this. We used a popular approach which proposes that high-motion frames ($FD > 0.40$) should be "scrubbed" prior to analysis, by replacing them with the mean of surrounding volumes. We have now added this clarification to the methods:

"8) scrubbing - using a frame-wise displacement threshold of 0.4. The maximum number of scrubbed volumes per scan was 7.1%; scrubbed volumes were replaced with the mean of the preceding and following volumes;"

Reviewer #4 (Remarks to the Author):

Singleton et al. present a compelling empirical evidence showing that (classical) psychedelics flatten the energy landscape of brain dynamics resulting in an agile transition between multiple brain states. This shows the brain's flexibility to visit (or revisit) various states under psychedelics hence demonstrating evidence for the therapeutic benefits these drugs can provide for people with internalising disorders which are marked by the trapping / rigidity to be stuck in a certain brain state. I didn't review the previous submission but I see that reviewers provide an extensive feedback which has resulted in a substantially improved paper especially in terms of the replication on two separate datasets one with psilocybin and other with music listening component. In that sense authors have done a very good job.

However, I do have one concern which wasn't raised in the previous round of review. This is regarding the rather amusing twist of the technical terminology. The RHEBUS hypothesis, which this paper is claiming to test, is specifically about the flattening of the "free" energy landscape while the current manuscript dropped "free" and talks about flattening of energy landscape. While this seems innocuous enough to a cursory reader however for a more technically minded readers there is clear discrepancy. Free energy is an information theoretic construct (also referred to as 'surprise' within the FEP framework), formally defined as the lower bound on the (log)model evidence. This reflects a cost function which is optimised when fitting a (generative) model to observed data. Whereas the (transition) energy metric used in the current manuscript is an entirely different construct defined as the "minimum energy required for the brain to transition from one activation pattern

to another". The RHEBUS paper (I am sorry that I need to describe it here for the author) combined entropic brain hypothesis and FEP which are both information theoretic arguments. An excerpt from RHEBUS: "The entropic brain hypothesis and free-energy principle are inter-related, not least because of their shared appeal to information theoretical metrics, closely linked to classic (Shannon) entropy:" and "The entropic brain measures the uncertainty of neuronal fluctuations across time, whereas free-energy measures the uncertainty of beliefs encoded by neuronal fluctuations." Current paper claims to test RHEBUS but then goes on to test a different quantity that was never mentioned in that paper which only talked about free energy. Free energy in RHEBUS and energy metric used in this paper to me are chalk and cheese and at best be only colloquially compared perhaps in the Discussion section. Transition energy as a proxy for free energy is a stretch too far. The only way I see that RHEBUS can be tested is using some constructs that appeal to model optimization and comparison (as is done in DCM). I would suggest that authors either find a way to resolve this technical discrepancy (which I am not sure is possible as this would mean going back in time and changing the RHEBUS hypothesis) or don't emphasise that this paper is testing RHEBUS which is more tenable position in my opinion. The presented evidence is very compelling on itself, replicated across datasets and can be guided by a rich literature on analysis of energy landscape as measure by fMRI (dozens of papers show up on the topic without a recourse to free energy optimization). I would suggest rewriting and repositioning of the current manuscript such that claims of testing RHEBUS (which is tied to testing hypothesis based on free energy) are not front and centre (probably only warranted in Discussion where speculative connections can be made).

We thank the reviewer for the thoughtful assessment of our manuscript. We have provided significant revisions to the manuscript that we hope will address this comment. In general, we have removed claims of validating a central tenet of REBUS (namely, the free energy portion), and instead focus on our own hypotheses that (a) control energy would be reduced under psychedelics, and (b) that this would relate to the increased entropy in brain activity. In the Limitations section, we clarify our position that reduced control energies may be a result of the relaxed priors and corresponding flattened free energy landscape hypothesized by REBUS, but do not claim to formally connect the two here, rather offering it as an avenue for future research:

"It is also not a direct measure of the brain's variational free energy landscape⁶⁶, an information theoretic topic foundational to REBUS. We hypothesize that the empirical changes in control energy demonstrated here indicate a flattening of the posterior, which may or may not arise from a relaxation of prior beliefs and a flattening of the free energy landscape as posited by REBUS. Future work will be required to determine if any direct relationship exists between free energy and control energy."

Reviewer #5 (Remarks to the Author):

None.

Comments from the editor, based on R5's comments to the editor:

-Please ensure abstract does not suggest PET data was generated under influence of hallucinogens

-Please discuss limitations of small sample size especially in relation to inferring different brain states and how this might affect the different findings for psilocybin and LSD trials in figure 3.

We have added emphasis throughout the manuscript that both the PET and SC data were obtained from publicly available sources and under non-drug conditions. As for Figure 3, we do already note in the results section that the difference in the findings may arise from the drugs themselves or the timeline of administration (“peak” versus onset). We have moved a sentence from the Results to the Limitations section where we discuss sample size and generalizability:

“Although the LSD dataset is more suitable for obtaining robust measurements of brain dynamics using the methods presented here, primarily because of the longer scans, and, in addition, the larger sample size, we still found that psilocybin modified the brain’s energy landscape and dynamics in a similar fashion to LSD.”

We have added a new sentence that follows the one above:

“Future experiments will be necessary to determine if the subtle differences in results found between LSD and psilocybin are a result of sample size, scan length, or pharmacodynamics (different receptor profiles, metabolism, or “peak” versus onset effects).”

Reviewers' comments:

Reviewer #1 (Remarks to the Author):

The authors have fully addressed my concerns. This is a nice contribution to the literature. Kudos.

Reviewer #4 (Remarks to the Author):

I do not have any further concerns and paper can be accepted.

REVIEWERS' COMMENTS

Reviewer #1 (Remarks to the Author):

The authors have fully addressed my concerns. This is a nice contribution to the literature. Kudos.

Reviewer #4 (Remarks to the Author):

I do not have any further concerns and paper can be accepted.

We would like to thank both reviewers for their thoughtful and thorough feedback on our manuscript.